# Assimilating solar-induced chlorophyll fluorescence into the terrestrial biosphere model BETHY-SCOPE v1.0: Model description and information content

Norton Alexander J.[1], Rayner Peter J.[1], Koffi Ernest N.[2], and Scholze Marko[3]

[1]School of Earth Sciences, University of Melbourne, Australia
[2]European Commission Joint Research Centre, Ispra, Italy
[3]Department of Physical Geography and Ecosystem Science, Lund University, Sweden

*Correspondence to:* A.J. Norton (nortona@student.unimelb.edu.au)

**Abstract.** The synthesis of model and observational information using data assimilation can improve our understanding of the terrestrial carbon cycle, a key component of the Earth's climate-carbon system. Here we provide a data assimilation framework for combining observations of solar-induced chlorophyll fluorescence (SIF) and a process-based model to improve estimates of terrestrial carbon uptake, or gross primary production (GPP). We then quantify and assess the constraint SIF provides on the uncertainty of global GPP through model process parameters in an error propagation study. By incorporating one year of SIF observations from the GOSAT satellite, we find that the parametric uncertainty in global annual GPP is reduced by 73%, from $\pm$ 19.0 $\mathrm{PgCyr^{-1}}$ to $\pm$ 5.2 $\mathrm{PgCyr^{-1}}$. This improvement is achieved through strong constraint of leaf growth processes and weak to moderate constraint of physiological parameters. We also find that the inclusion of uncertainty in shortwave down radiation forcing has a net-zero effect on uncertainty in GPP when incorporated in the SIF assimilation framework. This study demonstrates the powerful capacity of SIF to reduce uncertainties in process-based model estimates of GPP and the potential for improving our predictive capability of this uncertain carbon flux.

## 1 Introduction

The productivity of the terrestrial biosphere forms a key component of Earth's climate-carbon system. Estimates show that the terrestrial biosphere has removed about one quarter of all anthropogenic $CO_2$ emissions thus preventing additional climate warming (Ciais et al., 2013). Much of the interannual variability in atmospheric $CO_2$ concentration is also driven by terrestrial productivity. Despite this significance, understanding of the underlying mechanisms of terrestrial productivity is still lacking. This results in large uncertainties in the predictive capability of terrestrial productivity, and thus future predictions of atmospheric $CO_2$ and temperature (Friedlingstein et al., 2006).

A key challenge is disaggregating the observable net $CO_2$ flux into its component fluxes, gross primary production and ecosystem respiration. Gross primary production (GPP) is the rate of $CO_2$ uptake through plant photosynthesis and the largest natural surface to atmosphere flux of carbon on Earth (Ciais et al., 2013). Estimating spatiotemporal patterns of GPP at the scales required for global change and modeling studies has proven difficult. This is primarily due to two reasons, the complexity

of the processes involved and the difficulty in observing those processes (Baldocchi et al., 2016; Schimel et al., 2015). Remote sensing observations of solar-induced chlorophyll fluorescence (SIF) offer a novel constraint on GPP and the potential to partly address these two issues (Schimel et al., 2015).

At the leaf scale chlorophyll fluorescence is emitted from photosystems I and II during the light reactions of photosynthesis. These photosystems are pigment-protein complexes that form the reaction centers for converting light energy into chemical energy. It is in photosystem II (PSII) where photochemistry, the process initiating photosynthetic electron transport and leading to $CO_2$ fixation, is initiated. The link between chlorophyll fluorescence and photochemistry is confounded by a third key process however, heat dissipation, also termed non-photochemical quenching (NPQ). Both photochemistry and NPQ are regulated processes, responding to changing physiological and environmental conditions (Porcar-Castell et al., 2014). Changes in the rates of photochemistry and NPQ, and electron sinks other than $CO_2$ fixation, lead to a non-trivial, but direct link between chlorophyll fluorescence and photosynthetic rate (Flexas et al., 1999; Magney et al., 2017). Because chlorophyll fluorescence is tied in with these physiological processes it has become a highly useful indicator of the physiological state of leaves (see reviews by Baker, 2008; Porcar-Castell et al., 2014).

At the canopy scale and beyond the link appears simpler, exhibiting ecosystem-dependent linear relationships (Guanter et al., 2013). The slope of this linear relationship can change as the light-use efficiency of either SIF or GPP changes, for example due to water stress (Daumard et al., 2010) or changing light conditions (Yang et al., 2015). SIF also seems to outperform traditional remote sensing methods, such as Normalized Difference Vegetation Index (NDVI) and the Enhanced Vegetation Index (EVI) that use reflectance to derive vegetation indices, in tracking changes in GPP at this scale (Yang et al., 2015; Walther et al., 2016). This is in part because the SIF emission originates exclusively from plants, thus the retrieval is not contaminated by background materials like soil or snow. It is expected, however, that complicating factors such as the retrieval wavelength, temporal scaling, chlorophyll content, 3-dimensional canopy structure, and stress will also play a role in the GPP-SIF link (Damm et al., 2015; Guanter et al., 2012; Rossini et al., 2015; Zhang et al., 2016). Using high-resolution spectrometers onboard satellites global maps of SIF have been produced. A number of existing (GOME-2, GOSAT, OCO-2, TROPOMI, SCHIAMACHY) and planned (FLEX, GEOCARB) satellite missions are capable of measuring SIF. Utilizing these remotely-sensed SIF observations directly to track changes in GPP have already proven useful even without the addition of ancillary data or model information (Lee et al., 2013; Parazoo et al., 2013; Walther et al., 2016; Yang et al., 2015).

Data assimilation enables the use of observations and model information together to produce a best estimate of the state and function of the system. In the case of mechanistic models this is done by constraining the simulated processes and their parameters. Such an approach has been applied to terrestrial biosphere models to optimize model parameters and constrain the uncertainty in carbon flux estimates in a number of studies (see Kaminski et al., 2013; Koffi et al., 2013; Macbean et al., 2016; Peylin et al., 2016). The Carbon Cycle Data Assimilation System (CCDAS) is one such system and it has ingested observations such as atmospheric $CO_2$ concentration and/or the fraction of absorbed photosynthetically active radiation (FAPAR), demonstrating the benefit of combining model and observations in a regularized approach (Rayner et al., 2005; Kaminski et al., 2012). The use of SIF observations within a data assimilation framework may provide a highly useful, complementary constraint on GPP. While one study by Parazoo et al. (2014) utilized SIF in a data assimilation system to redistribute multiple model es-

timates of GPP, no optimization of model process parameters was performed. Koffi et al. (2015) incorporated a mechanistic model for SIF into the CCDAS system and then conducted sensitivity tests and compared model simulated SIF and observed SIF from GOSAT demonstrating the model is capable of ingesting the data. However, SIF has not yet been used on a global scale in a data assimilation system to optimize process parameters.

In this paper, we assess the ability of satellite SIF observations to constrain the parametric uncertainty of simulated GPP in a terrestrial biosphere model within a data assimilation system. This is termed an error propagation study and is similar in concept to an observing system simulation experiment or quantitative network design study (Hungershoefer et al., 2010; Kaminski et al., 2010; Koffi et al., 2013). Parameters and simulated GPP are therefore optimized only for their uncertainty and not for their absolute quantities. Considering SIF is a novel observational constraint, this is an important first step toward a full

assimilation of the data as it allows us to test whether an assimilation of SIF data will be beneficial for reducing uncertainty in GPP. This is performed by estimating the constraint that SIF provides on the uncertainty of model parameters and the parametric uncertainty of model simulated GPP.

## 2   Methods

Under the linear Gaussian assumption, the uncertainty of a target quantity (here, GPP) following assimilation of the data (here,

SIF) is conditional only on the prior uncertainty, the uncertainty of the observations and the sensitivity of simulated observations to changes in the parameters (Tarantola, 2005). Given we apply this assumption to estimate posterior uncertainties, this linear problem can be performed independently of the optimization of the parameter values. The model used for determining the sensitivity of simulated observations to changes in the parameters is run at a relatively low spatial resolution which provides high computational efficiency. We note that subsequent work to assimilate the data should be performed at a higher spatial

resolution in order to better represent the heterogeneity of the land surface.

    We formulate this error propagation study into two stages: (i) optimization of parameter uncertainties, and (ii) projection of the parameter uncertainties onto uncertainty in diagnostic GPP. Here, we outline the model used to simulate the observation (SIF) and the target quantity (GPP). We also outline the model parameter set describing these processes, the uncertainty in the observations and model forcing, and general experimental setup.

**2.1   Model Description**

In order to ingest an observation into a data assimilation system, we require a model or 'observation operator' that can simulate SIF, ideally providing a process-based relationship between SIF and GPP. There are a few ways one might formulate the observation operator. Evidence shows a strong linear relationship between SIF and GPP at large spatial scales and relatively long temporal scales (Frankenberg et al., 2011b; Guanter et al., 2012), suggesting relatively simple scaling between GPP and

SIF. However, it is known that the link is more complex than this, and it is expected to differ at finer spatial and temporal scales due to, for example, land surface heterogeneity or the time of day of the measurements. To ensure the model has these capabilities we have opted for a process-based observation operator.

In this section we describe the newly developed terrestrial biosphere model for simulating and assimilating SIF. The model is an integration of the existing models BETHY (Biosphere Energy Transfer Hydrology) (Rayner et al., 2005; Knorr et al., 2010) and SCOPE (Soil Canopy Observation, Photosynthesis and Energy fluxes) (Van der Tol et al., 2009) and builds upon the developments of Koffi et al. (2015). The coupling of BETHY and SCOPE enables spatially explicit, plant-type dependent,
global simulations of GPP and SIF. This model may be run on a computationally efficient, low spatial resolution grid of 7.5° × 10° or a high spatial resolution grid of 2° × 2°.

BETHY is a process based terrestrial biosphere model at the core of the Carbon Cycle Data Assimilation System (CCDAS) (Rayner et al., 2005; Scholze et al., 2007). Full model description details can be found elsewhere (e.g. Rayner et al., 2005; Scholze et al., 2007; Knorr et al., 2010). Briefly, BETHY simulates carbon assimilation and plant and soil respiration within
a full energy and water balance. The version used here also incorporates a leaf area dynamics module for prognostic leaf area index (LAI) as described in Knorr et al. (2010). This module includes parameters for leaf development, phenology and senescence processes (hereby collectively termed leaf growth) to determine LAI in a scheme that incorporates temperature, water and light limitations on growth and is capable of representing the major global phenology types (Knorr et al., 2010). This scheme also enables the representation of subgrid variability in leaf growth, representing the likely variability in growth
triggers across a grid cell and the necessary form for differentiability between process parameters and state variables. The full BETHY model consists of four key modules: (i) energy and water balance; (ii) photosynthesis; (iii) leaf growth and; (iv) carbon balance. It represents variability in physiology and leaf growth of plant classes by 13 plant functional types (PFTs) (see Table 1) originally based on classifications by Wilson and Henderson-Sellers (1985). Each model grid cell may consist of up to three PFTs as defined by their grid cell fractional coverage.
SCOPE is a vertical (1-D) integrated radiative transfer and energy balance model with modules for photosynthesis and chlorophyll fluorescence (Van der Tol et al., 2009). At present it is the only process-based model capable of simulating canopy-scale chlorophyll fluorescence. SCOPE incorporates current understanding of chlorophyll fluorescence processes including canopy radiative transfer, re-absorption of fluorescence within the canopy, and the non-linear relationship between chlorophyll fluorescence quantum yield and other quenching processes (Van der Tol et al., 2009, 2014). Leaf level chlorophyll fluorescence
is coupled to the commonly used Farquhar and Collatz models for C3 and C4 photosynthesis, respectively (Van der Tol et al., 2009). A current limitation of SCOPE is that there is no link between leaf level biochemistry and soil moisture. This is partly compensated by changes in LAI due to soil moisture as simulated by BETHY.

The canopy radiative transfer and photosynthesis schemes of BETHY have been replaced by the corresponding schemes in SCOPE, including the components required for calculation of chlorophyll fluorescence at leaf and canopy scales. The spatial
resolution, vegetation (PFT) characteristics, leaf growth, and carbon balance are handled by BETHY. SCOPE therefore takes in climate forcing (meteorological and radiation data) and LAI from BETHY, and returns GPP. BETHY calculates the canopy water balance, leaf growth, and net carbon fluxes, which will prove useful in future when assimilating other data streams (e.g. atmospheric $CO_2$ concentration). Importantly, SCOPE provides a process-based link between SIF and GPP allowing the transfer of information from observations of SIF to simulated GPP. Subsequently, information from SIF may also be transferred
to carbon fluxes resulting from GPP such as net ecosystem productivity.

**Table 1.** PFTs defined in BETHY and their abbreviations.

| PFT # | PFT Name | Abbreviation |
|-------|----------|--------------|
| 1 | Tropical broadleaved evergreen tree | TrEv |
| 2 | Tropical broadleaved deciduous tree | TrDec |
| 3 | Temperate broadleaved evergreen tree | TmpEv |
| 4 | Temperate broadleaved deciduous tree | TmpDec |
| 5 | Evergreen coniferous tree | EvCn |
| 6 | Deciduous coniferous tree | DecCn |
| 7 | Evergreen shrub | EvShr |
| 8 | Deciduous shrub | DecShr |
| 9 | C3 grass | C3Gr |
| 10 | C4 grass | C4Gr |
| 11 | Tundra vegetation | Tund |
| 12 | Swamp vegetation | Wetl |
| 13 | Crops | Crop |

## 2.2 Model Process Parameters

In this error propagation system, information from the SIF observations is used to constrain the uncertainty of the model process parameters. Parameters can either be global or differentiated by PFT. Global parameters apply to plants or soils everywhere while PFT-dependent parameters enable differentiation between physiological and leaf growth traits. Some key parameters

for this study such as the maximum carboxylation capacity ($V_{cmax}$) and chlorophyll $a/b$ content ($C_{ab}$) are considered PFT-dependent. From an ecophysiological perspective, there are other parameters specific to SCOPE that may be considered PFT-dependent such as the vegetation height and leaf angle distribution parameters. However, we have assumed them to be global to simplify the problem. GPP is relatively insensitive to these parameters, so this is not expected to impact the GPP uncertainty reduction results. Despite this, in a full assimilation with the SIF data it may be necessary to make these PFT-dependent to

improve the model-observed fit.

We expose 53 parameters from BETHY-SCOPE to the error propagation system (see Table A1). As stated above, each of these is represented by its PDF, assumed to be Gaussian. The mean and standard deviation for the prior parameters is shown Table A1. Choice of the prior mean and uncertainty for parameters follow those used in previous studies (Kaminski et al., 2012; Knorr et al., 2010; Koffi et al., 2015). For new parameters that are not well characterized (e.g. SCOPE parameters) we assign

relatively large prior uncertainties, and mean values in line with the default SCOPE parameters and with Koffi et al. (2015). The choice of the prior may be considered important here considering we are using a linear approximation of the model around

$x_0$ and that the model is known to be non-linear. Therefore, sensitivities can differ depending upon the choice of $x_0$ (Koffi et al., 2015).

There are twelve SCOPE parameters exposed, one of which is PFT-dependent. These parameters were chosen due to their importance in simulating SIF or GPP, and to sensitivity tests such as those performed by Verrelst et al. (2015). They include
$C_{ab}$, leaf dry matter content ($C_{dm}$), leaf senescent material fraction ($C_s$), two leaf distribution function parameters ($LIDFa$, $LIDFb$), vegetation height ($hc$) and leaf width. Leaf physiological parameters include $V_{cmax}$, Michaelis-Menten kinetic coefficients for $CO_2$ ($K_C$) and $O_2$ ($K_O$), the ratio of the Rubisco oxygenation rate to $V_{cmax}$ ($\alpha_{V_o,V_c}$), and the ratio of day respiration to $V_{cmax}$ ($\alpha_{R_d,V_c}$).

## 2.3 Uncertainty Calculations

To calculate the uncertainty in parameter values following the constraint provided by the observational information of SIF (i.e. the posterior uncertainty) we propagate uncertainty from the observations onto the parameters. In order to perform this, we utilize a probabilistic framework where the state of information on parameters and observations is expressed by their corresponding probability density functions (PDF) (see Tarantola, 2005). The probability density of the errors in these quantities is assumed to be Gaussian, thus they are describable by their mean and uncertainty. The prior information on parameters is
quantified by a PDF in parameter space and the observational information by a PDF in observational space. The mean values for the parameters and observations are denoted by $x$ and $d$, respectively. The uncertainty covariance matrices in parameter space and observational space are denoted by $C_x$ and $C_d$, respectively.

For linear and weakly non-linear problems we can assume that Gaussian probability densities propagate forward through to Gaussian distributed simulated quantities (Tarantola, 2005). This permits linear error propagation from the input parameters to
the model outputs. Estimating posterior uncertainties of the parameters for these types of problems can therefore be performed independently of the parameter estimation, in other words without the need to constrain the mean values of the parameters (Kaminski et al., 2010, 2012). This requires a matrix of partial derivatives of a target quantity with respect to its variables, also called a Jacobian matrix ($H$). This matrix represents the sensitivity of a simulated quantity (e.g. SIF, GPP) to the parameters. With the linear approximation, $H$ is calculated around the prior parameter values ($x_0$). This simplification of the model sen-
sitivity brings limitations to the accuracy of the method. However, with the aggregation of subgrid variability across a model grid cell, sudden shifts in model sensitivity (e.g. step functions) are less likely or realistic; the present model incorporates these effects (Knorr et al., 2010). Additionally, because the parameter space can be very large, the use of prior knowledge on $x_0$ helps to limit the effect of this problem as $H$ at $x_0$ likely provides a decent approximation of the true $H$ that would occur at the global optimum (Tarantola, 2005). The simplification is also useful considering the high computational cost of calculating $H$.
To calculate the posterior parameter covariance matrix ($C_{x_{post}}$) following constraint by observational information, $C_d$, we use Eq. 1 (Tarantola, 2005).

$$C_{x_{post}}^{-1} = C_{x_0}^{-1} + H^T C_d^{-1} H \tag{1}$$

Where $H$ expresses the Jacobian for SIF and $H^T$ the Jacobian transposed. Comparing parameter uncertainties in the prior ($C_{x_0}$) and the posterior ($C_{x_{post}}$) allows us to quantify the improvement in parameter precision following the observational constraint. The parameter uncertainties in $C_{x_0}$ and $C_{x_{post}}$ may be expressed as standard deviations ($\sigma$) by calculating the square root of their diagonal elements. We can therefore assess the relative uncertainty reduction in parameter following SIF constraint, or 'effective constraint', with 1 - ($\sigma_{posterior}/\sigma_{prior}$). This quantifies the effective constraint of the prior uncertainty and may be represented as a percentage decrease in $\sigma$ uncertainty.

Formally, $C_d$ represents the errors in the measurements and in the model simulated counterpart (i.e. model error) (Scholze et al., 2016). As described further below in section 2.4, we only consider the contribution of measurement errors to $C_d$ in calculating posterior probabilities. However, to see if the assumptions that we have made about uncertainties are consistent with the model-data mismatch, we assess the reduced $\chi^2$ statistic ($\chi_r^2$) similar to a method employed by Kuppel et al. (2013). While more formal approaches to optimally estimate covariance parameters exist (Michalak et al., 2005), this metric can highlight whether we are neglecting a significant source of error in $C_d$, for example, model structural error. It also provides an indication of whether the model is capable of reproducing the measurements, given the assumed uncertainties. This is calculated by

$$\chi_r^2 = \frac{1}{N}\big(M(x_0) - d\big)\big(HC_{x_0}H^T + C_d\big)^{-1}\big(M(x_0) - d\big) \qquad (2)$$

where $N$ is the number of degrees of freedom (equal to the number of observations in this case), $M(x_0)$ is the forward model simulated SIF for the prior case, and $d$ is the SIF observations. A $\chi_r^2$ greater than one would indicate that our assumptions around uncertainties may not be valid given the model-data mismatch and that the model cannot simulate the measurements (Michalak et al., 2005). Conversely, a $\chi_r^2$ of less than one indicates over-confidence in the assumed uncertainties. A value of approximately one is most desirable as it would indicate that our overall assumptions of uncertainties are valid. At the low-resolution applied in the information content analysis, representation errors will be relatively large and may dominate other sources of error in Eq. 2 and therefore mask the actual ability of the model to simulate the measurements. For an assimilation of the data, the model would not be used at such a low resolution given the heterogeneity of the land surface and would instead be run at a higher spatial resolution. To help reduce the representation error we utilise unpublished work that compares the forward model at a higher resolution ($2° \times 2°$) and with SIF observations from the OCO-2 satellite for 2015. While this uses a slightly different parameterization, it is more credible and helps minimise the effects of representation error in determining whether the model can simulate the measurements. The error propagation analysis, however, benefits from using the low-resolution as it greatly improves computational efficiency considering the computational demand of the model simulations and subsequent calculations.

The observational constraint introduces correlations into the posterior parameter distributions, thus posterior parameter uncertainties are not wholly independent. Strong correlations in $C_{x_{post}}$ indicate parameters that cannot be resolved independently

in an assimilation, however their linear combinations can be. We calculate correlations in parameters by expressing the covariances as correlations as in Eq. 3 (see Tarantola, 2005, p.71) by

$$R_{i,j} = \frac{C_{i,j}}{\sqrt{C_{i,i}}\sqrt{C_{j,j}}} \tag{3}$$

where diagonal elements have a correlation equal to one while off-diagonals elements can range between -1 and 1. If large enough, these correlations can contribute significantly to the overall constraint of the target quantity (Bodman, 2013).

Using the parameter covariance matrix we can assess how parameter uncertainties propagate forward through the model onto uncertainty in GPP using the Jacobian rule of probabilities, the same method outlined in Rayner et al. (2005). This is the second stage of our error propagation study. Using $C_{x_0}$ we estimate the prior uncertainty in a vector of simulated target quantities (i.e. GPP). Similarly, using $C_{x_{post}}$ we estimate the posterior uncertainty in a vector of simulated target quantities. We calculate the uncertainty covariance of GPP ($C_{GPP}$) using Eq. 4.

$$C_{GPP} = H_{GPP}C_x H_{GPP}^T \tag{4}$$

Where $H_{GPP}$ is the Jacobian matrix of GPP with respect to the parameters. With this we can quantify the improvement in precision of simulated GPP by using either $C_{x_0}$ or $C_{x_{post}}$ in Eq. 4. Therefore, using the forward model, a statistical estimation scheme and a set of observational uncertainties we can assess the information content of the SIF observations in the context of the model, its parameter set, and simulated GPP taking explicit consideration of uncertainties.

## 2.4 Uncertainty in Observations and Model Forcing Variables

The uncertainty of the measured data (hereafter, data) is a critical component in assessing the potential impact of an observing system on the estimation of carbon fluxes. Data uncertainties in SIF used here are calculated from the GOSAT satellite observations for 2010. This data is obtained from the ACOS (Atmospheric $CO_2$ Observations from Space) project at a grid resolution of 3° × 3°. As the model simulations are performed on a low-resolution grid (7.5° × 10°), we aggregate these uncertainties to this resolution using Eq. 5 as described below in a way that conserves the information content from the original 3° × 3°observations.

We assume the observations are independent and have uncorrelated errors, that is, they are distributed randomly. Assuming uncorrelated errors is, however, likely to overestimate the information content particularly if using the standard error as the uncertainty. Although it has been used in recent studies with satellite SIF (e.g. Parazoo et al., 2014), the standard error under an assumption of uncorrelated errors is likely to be an overly optimistic approximation of the information content. For this study, we take a slightly conservative approach, scaling the calculated standard error by the square root of two as shown in Eq. 5. This effectively doubles the variance in an independent dimension and reduces the information content to compensate for the assumption of uncorrelated errors.

Through aggregation of GOSAT grid cells to the model grid resolution the number of independent measurements is reduced. To account for this and preserve the information content of the original GOSAT observations the uncertainty in a given model grid cell is, approximately, divided by the square root of the number of GOSAT grid cells with SIF data that fall within that model grid cell ($N$). More precisely, we apply an area-weighting term in the equation (see Supplementary material Eq. A1). This has the effect of scaling the uncertainty by the $1/\sqrt{N}$ law, but takes into account the fact that SIF is in physical units per units area (i.e. $W\ m^{-2}\ \mu m^{-1}\ sr^{-1}$) and that grid cells have different areas over different latitudes. A full description of this calculation and detailed example is shown in supplementary material.

Therefore, the calculation of the SIF data uncertainties used here is approximated by Eq. 5 (for further details see Supplementary material Section A2). For a given model grid cell, the variance ($\sigma^2$) is approximately equal to the sum of the standard error of each individual GOSAT grid cell ($\sigma_i$) squared, then scaled by the number of individual GOSAT grid cells with data and the square root of two.

$$\sigma^2 = \sqrt{2} \left[ \frac{1}{\sqrt{N}} \sum_i \sigma_i^2 \right] \tag{5}$$

The resulting annual observational uncertainties, shown in Figure 3, appear to be much smaller than the uncertainties of individual GOSAT grid cells. In part this is due to the aggregation of multiple independent observations. Regions with more soundings across the year (e.g. the tropics) will also have smaller annual uncertainties.

Uncertainty of SIF observations may also have a systematic component. A known, potential systematic error in SIF stems from the zero-level calculated during the retrieval. Any error in the calculated zero-level offset will add to the measurement error. This radiometric correction is done to prevent biases in the SIF retrieval (Frankenberg et al., 2011a; Guanter et al., 2012) and this is performed monthly in the present GOSAT retrieval of SIF. In this case, it is systematic in the sense that it applies to multiple measurements. This type of error is distinguished from a bias which is a systematic error with a precisely known magnitude and sign that should be corrected for. A bias cannot be incorporated into the present error progation framework whereas an error in the zero-level offset can be provided it is Guassian. To clarify, a retrieved measurement ($d$) of a quantity (e.g. SIF) at index point $i$ can be given by

$$d_i = d_i^t + \varepsilon_i + \varepsilon_z \tag{6}$$

where $d_i^t$ is the true value at index point $i$, $\varepsilon_i$ is a random variable with a variance of $\sigma^2$ at index point $i$, and $\varepsilon_z$ is a random variable that has some variance and is constant for a subset of the measurements (e.g. across a particular region or time). Based on previous analyses of the instruments, the error in zero-level offset in the SIF retrieval may be considered small (Frankenberg et al., 2011a, 2014). Here, we provide a more detailed assessment and characterization of the in-orbit systematic error. This is performed by assessing zero-level offset corrected GOSAT SIF soundings over the non-fluorescent regions of Antarctica and central Greenland during January and July, respectively (see Appendix Figure 10), in order to sample the error distribution of $\varepsilon_z$. These systematic errors appear quite small ($\pm\ 0.06\ W\ m^{-2}\ \mu m^{-1}\ sr^{-1}$) and may vary seasonally due to factors such

as atmospheric conditions or instrument-related causes (Guanter et al., 2012). We therefore assess the effect of a conservative systematic random error of size $\pm\,0.1\;W\;m^{-2}\;\mu m^{-1}\;sr^{-1}$ in the zero-level offset seasonally. Practically, this means adding four (one for each season) extra uncertainty terms to $C_x$ corresponding the the estimated error and adding four extra terms in $H$ which are scaling terms (equal to one) applied to the corresponding season. Including these terms provides a sensitivity test to indicate how an error in the zero-level offset propagates through to uncertainty in GPP.

An additional source of uncertainty in model estimates of GPP is climate forcing. As mentioned by Koffi et al. (2015), while uncertainty in forcing such as incoming radiation is not considered in the current CCDAS setup, it is considered to be an important variable in driving SIF (Verrelst et al., 2015) and GPP (Farquhar et al., 1980). Without consideration of uncertainties in forcing variables the uncertainty in GPP may be underestimated. Studies that use process-based models or empirically-derived relationships do not explicitly consider such uncertainties (e.g. Beer et al., 2010). One such forcing variable

is downward shortwave radiation (SWRad). Monthly means of SWRad are suggested to have a random error of 12 $Wm^{-2}$ (6% of the mean) due mostly to uncertainty in clouds and aerosols (Kato et al., 2012). We therefore investigate how this random error in SWRad may be considered in GPP estimates. Furthermore, as SIF responds strongly to SWRad, there is the potential to utilize SIF observations as a constraint on the uncertainty of the forcing. We therefore conduct an additional experiment that incorporates the uncertainty in SWRad in the error propagation system. For this experiment an additional

parameter representing SWRad is added to the inversion, which acts as a scaling factor for SWRad globally. We investigate the level of constraint SIF provides on this scaling factor, and the subsequent effects of incorporating uncertainty in SWRad in this inversion on uncertainty in GPP.

## 2.5   Model and Data Setup

In this study BETHY-SCOPE is run for the year 2010 on the computationally efficient, low-resolution spatial grid (7.5° ×

10°). As the dynamical equations are the same for either low-resolution or high-resolution scales, use of the low-resolution setup is appropriate for an error propagation study as long as careful consideration is taken with observational uncertainties. Climate forcing in the form of daily meteorological input fields for running the model (precipitation, minimum and maximum temperatures, and incoming solar radiation) were obtained from the WATCH/ERA Interim data set (WFDEI Weedon et al., 2014). Photosynthesis and fluorescence are simulated at an hourly time step but forced by the respective monthly mean diurnal

cycle. Leaf growth and hydrology are simulated daily.

SIF is simulated at 755 nm, the wavelength corresponding to the GOSAT retrieval frequency and near to the OCO-2 retrieval frequency (757 nm). We focus upon the constraint by SIF measurements at 1:00 p.m. local time as it closely corresponds to the local overpass time of the SIF-observing satellites GOSAT and OCO-2. However, we also investigate the effect of using alternative SIF-observing times (e.g. the GOME-2 satellite overpass time) and multiple observing times simultaneously on the constraint of GPP.

## 3 Results

### 3.1 Prior Mismatch

First, we present the results from Eq. 2 that determines whether the assumed uncertainties allow for coverage of observed
SIF. As described in the methods, in this case we use a model forward run using the high-resolution version of the model and
compare this with a SIF observations from the OCO-2 satellite. We find that $\chi_r^2 = 0.97$ in this high-resolution case, close to the
optimal value of one.

### 3.2 Parameter Uncertainties

A key metric for assessing the relative uncertainty reduction, or 'effective constraint', is defined as $1-(\sigma_{posterior}/\sigma_{prior})$. The
effective constraint for all 53 parameters following constraint by SIF is shown in Figure 1 and in Table A1. We define weak,
moderate and strong effective constraint as the relative uncertainty reduction from 1-10%, 10-50%, and >50%, respectively.

Parameters describing leaf composition ($C_{ab}$, $C_{dm}$, $C_{sm}$) generally achieve strong effective constraint from SIF. For eleven
of the thirteen $C_{ab}$ parameters the uncertainty is strongly constrained, between about 50% and 85%. SIF is highly sensitive
to $C_{ab}$ and we assign a relatively large prior uncertainty on these parameters, so considerable constraint is expected. For
the tropical broadleaved evergreen tree PFT however, the effective constraint on $C_{ab}$ is much lower at 7%. For other leaf
composition parameters $C_{dm}$ and $C_{sm}$ SIF effectively constrains the uncertainty by 1% and <1% respectively.

Varied effective constraint is seen for the leaf growth parameters (parameters 18-34 in Table A1) that control phenology
and leaf area. Four out of the seventeen leaf growth parameters exhibit strong uncertainty reductions. These parameters pertain
to a variety of processes including the temperature at leaf onset, day length at leaf shedding, leaf longevity, and the expected
length of dry spell before leaf shedding ($\tau_W$) (see Table A1). The parameter $\tau_W$ is important in controlling leaf area and it
sees strong effective constraint from SIF, from 38-65% depending upon which class of PFT it pertains to. For the parameters
that are PFT-specific, there is generally a larger constraint seen when they relate to the C3Gr, C4Gr and crops. For example,
uncertainty in $\tau_W$ for grasses and crops ($\tau_W^{Gr}$) is effectively constrained by 65%.

Leaf physiological parameters (parameters 1-17 in Table A1) see a weak to moderate level of effective constraint. Of partic-
ular importance for simulating GPP is the PFT-specific parameter $V_{cmax}$. Effective constraint on $V_{cmax}$ varies from <1% up to
30% depending upon the PFT of interest. Five PFTs that, combined, represent about 65% of the land surface have their $V_{cmax}$
parameters constrained by >10%. The global physiological parameters include the ratio of the maximum rate of oxygenation
($V_{omax}$) to $V_{cmax}$ ($a_{V_o,V_c}$), the ratio of dark respiration ($R_d$) to $V_{cmax}$ ($a_{R_d,V_c}$), and the Michaelis-Menten enzyme kinetic
constants of Rubisco for $CO_2$ ($K_C$) and $O_2$ ($K_O$). These all see very weak effective constraint from SIF (<1%).

Global canopy structure parameters (parameters 50-53 in Table A1) also see a weak to moderate constraint from SIF. In
particular the structural parameters $LIDFa$ and $LIDFb$ see their uncertainty reduced by 22% and 9%, respectively. The
parameters for vegetation height and leaf width, which are used to calculate the fluorescence "hot-spot" variable (see Van der
Tol et al., 2009), are effectively constrained by 7% and <1%, respectively.

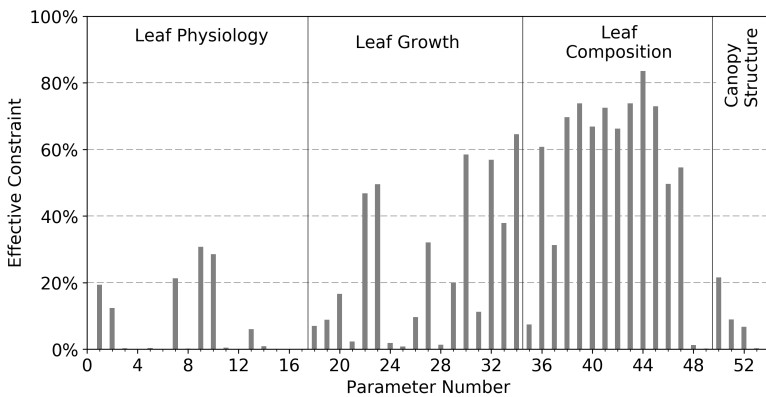

**Figure 1.** Effective constraint of BETHY-SCOPE model process parameters from SIF observations. Only the parameter numbers are given, for the corresponding descriptions see Table A1.

Parameters that pertain to more dominant PFTs in terms of land surface coverage (e.g. C3 grass) tend to see stronger uncertainty reductions. This is due to them being exposed to more SIF observations.

With the observational constraint correlations are introduced into the posterior parameter distributions. We assess these correlations using 3, shown in Figure 2. We find strong ($R \geq 0.5$) positive correlations between nine of the PFT-specific $C_{ab}$ parameters. These are also negatively correlated the leaf angle distribution parameter $LIDFa$. Thus, during a full assimilation with SIF data only the sum of $C_{ab}$ and $LIDFa$ can be resolved, not their individual values. Two leaf growth parameters are also strongly correlated, $T_\phi$ with $T_r$. Smaller correlations are also present between the subset of parameters shown in Figure 2.

To assess the effect of incorporating a systematic error from the observations into this analysis we apply a seasonal $\sigma$ error of 0.1 $W\ m^{-2}\ \mu m^{-1}\ sr^{-1}$ (equivalent to $\varepsilon_z$ in Eq. 6). This is incorporated as four additional parameters, one for each season, that scale the SIF signal across the globe. We find that the inclusion of this systematic error has a negligible effect on posterior uncertainties of the parameters. The difference in effective constraint between this sensitivity test case and the standard case above is <1% for any given parameter.

### 3.3 Uncertainty in GPP

To assess the constraint imposed by SIF on simulated GPP we compare the prior and posterior uncertainty in GPP as calculated using Eq. 4. Similar to the assessment of parameter uncertainty reductions, to assess the effective constraint of SIF on GPP we use a metric that measures the relative uncertainty reduction in $\sigma$ from the prior to the posterior.

Global GPP from the prior model is approximately 164 $\mathrm{PgCyr}^{-1}$ with a prior uncertainty of 19.0 $\mathrm{PgCyr}^{-1}$. Utilizing SIF observations at 1:00 p.m. results in a 73% reduction of the prior uncertainty giving a posterior of 5.2 $\mathrm{PgCyr}^{-1}$. Spatially, the prior uncertainty in GPP varies across the globe, with particularly large uncertainties in regions with high productivity (Figure 4). This is to be expected considering GPP uncertainty will typically correlate with absolute GPP. In the posterior, it is clear

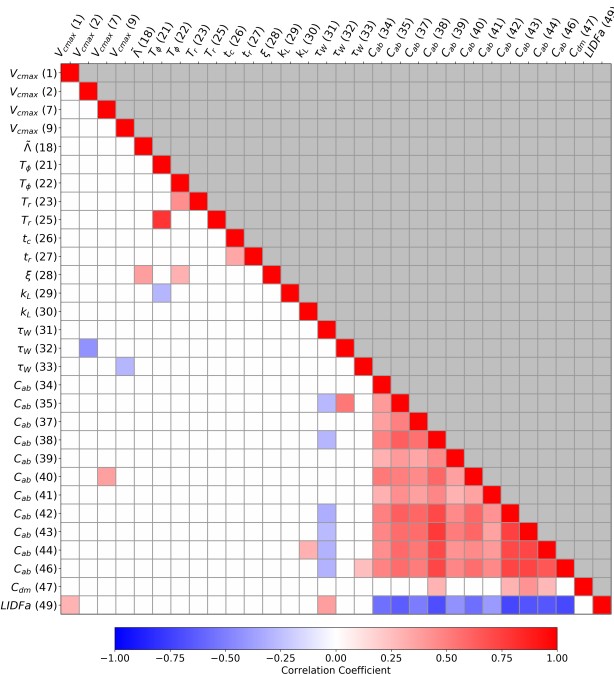

**Figure 2.** Correlation coefficients (r-value) in the posterior parameter covariance matrix ($C_{x_{post}}$). This shows the magnitude and sign of correlations in posterior parameter uncertainties following constraint with SIF data. Only parameters with an absolute correlation coefficient >0.25 with one or more other parameters are shown. Values above and below the diagonal are identical, therefore those above are coloured grey. The axes labels show the parameter symbol and number as defined in Table A1.

that uncertainty in GPP is strongly reduced across the globe (Figure 5). The relative uncertainty reduction (Figure 6) appears to show smaller constraint of uncertainty in the boreal regions, however this is because prior uncertainty is already relatively low (Figure 4).

5    To assess which parameters contribute to the uncertainty in GPP for the prior and posterior, we can conduct linear analysis of the uncertainty contributions. Typically this technique can only be used for the prior as the correlations in posterior parameter uncertainties, excluded from the linear analysis, also contribute toward the overall constraint. However, we can assess the contribution of these correlations to the constraint of GPP by setting the off-diagonal elements in $C_{x_{post}}$ to zero and using it in Eq. 4; the difference between this and the standard case that uses the full $C_{x_{post}}$ equates to the contribution of correlations. We

10    find that the contribution of these correlations to the constraint of GPP is small (0.16 $\mathrm{PgCyr}^{-1}$ or <1%), thus we can assume the linear analysis technique holds for the posterior as well. This finding is supported by the correlation analysis in posterior parameter uncertainties which showed few significant correlations in parameters relevant for GPP. This result is encouraging as it indicates that the parameters in a SIF assimilation system contributing most to the constraint of GPP are capable of being resolved independently.

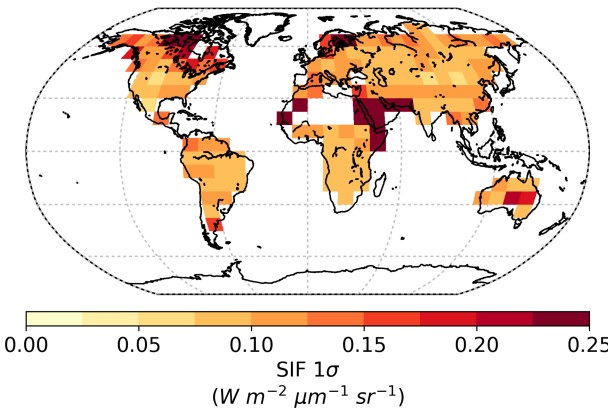

**Figure 3.** Annual observational uncertainty of SIF interpolated from GOSAT observations for 2010.

Using linear analysis of the uncertainty we find that uncertainty in global annual GPP in the prior and posterior stems from different processes. For the prior we see that the uncertainty in GPP is dominated, at 89%, by parameters describing leaf growth processes. Of these, a single parameter, $\tau_W$ for C3 grass, C4 grass and crops ($\tau_W^{Gr}$) makes up 74% of the uncertainty in global
annual GPP. Parameters representing physiological processes account for about 9% of prior uncertainty, most of which stem from the $V_{cmax}$ parameters. Parameters for $C_{ab}$ only account for 2.5% of the uncertainty.

For the posterior, which has a lower overall uncertainty in GPP, uncertainty is dominated by parameters representing physiological processes. Physiological parameters account for 67% of the uncertainty in posterior annual GPP, with $V_{cmax}$ parameters accounting for 32% and the Michaelis-Menten constant of Rubisco for $CO_2$ ($K_C$) accounting for 30%. The relative contribu-
tion by leaf growth parameters is reduced to 33%, and for $\tau_W^{Gr}$ to 15%. For $C_{ab}$ the relative contribution is smaller than the prior at <1%. This shift in which parameters contribute to the relative uncertainty in GPP between the prior and the posterior demonstrates how effectively SIF constrains leaf growth processes. Uncertainties in physiological parameters are constrained less than the leaf growth parameters which results in them contributing more in relative terms to the posterior uncertainty of GPP.

Regionally, we split the land into three regions, the Boreal region (above 45° North), the Temperate North (30° to 45° North) and the Tropics (30° South to 30° North). SIF constraint on annual GPP varies substantially across different regions of the globe, with relative uncertainty reduction in of 48%, 82%, and 79% for the Boreal, Temperate North and Tropics regions, respectively. In Figure 7 we show the contribution of parameter classes (leaf physiology, leaf growth, leaf composition and canopy structure; see Table A1 for details) to the parametric uncertainty of GPP across the year for each of these regions. From
Figure 7 it can be seen that the Boreal and Temperate North regions exhibit seasonal differences in total uncertainty and in the constraint SIF provides. This is caused by seasonal dependencies in the sensitivity of SIF and GPP to certain processes (e.g. leaf development versus leaf senescence) as well as seasonal differences in the density of observations in these regions. There

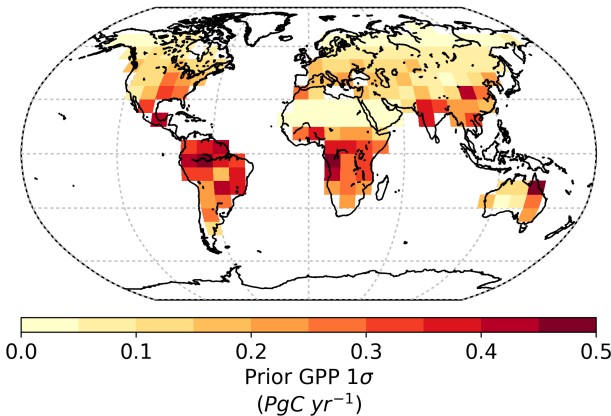

**Figure 4.** Prior parametric uncertainty in annual GPP.

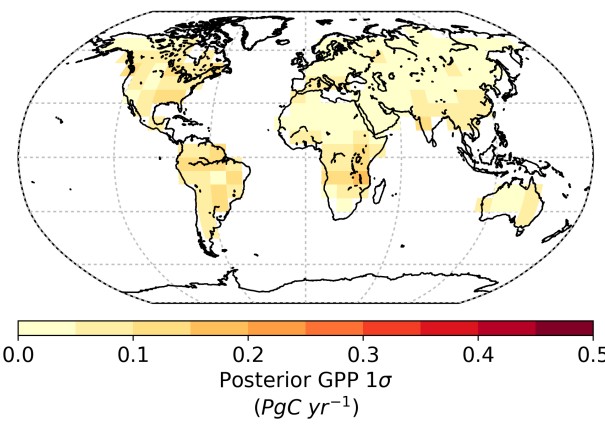

**Figure 5.** Posterior parametric uncertainty in annual GPP.

are far fewer GOSAT satellite observations during Boreal autumn and winter, thus there are fewer observations to constrain processes controlling GPP during this time.

During the start of the growing season leaf physiology, in particular photosynthetic rate constants ($V_{cmax}$), play a larger role whereas later in the growing season during the warmest months leaf growth, via water limitation on leaf area ($\tau_W^{Gr}$) of grasses, plays a larger role. Therefore in the Boreal region, where the strongest seasonality in constraint is seen, from July through to January SIF constrains GPP by >60%. Uncertainty in GPP during these months is dominated by the leaf growth parameters $\tau_W^{Gr}$ and $k_L$ along with $C_{ab}$ (for EvCn) all of which receive considerable constraint from SIF. From February to June however, SIF constrains GPP by less than 50%, as a large proportion of the uncertainty arises from the less-constrained $V_{cmax}$ parameters.

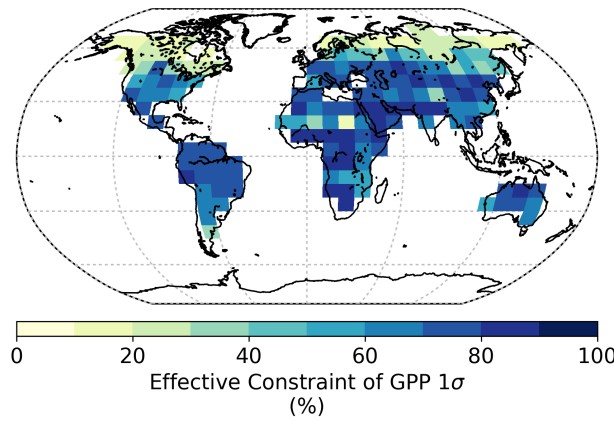

**Figure 6.** Relative uncertainty reduction (i.e. effective constraint) of parametric uncertainty in annual GPP from prior to posterior.

Following SIF constraint, uncertainty in Boreal GPP stems mostly from uncertainty in leaf physiology, particularly for the EvCn PFT. Similar differences between seasonal constraint is seen for the Temperate North, although with a smaller seasonal variation in SIF constraint that ranges between 74% and 87% across the year.

For the Tropics uncertainty reduction in GPP is about 80% across the year. Uncertainty in the prior is dominated by the leaf growth parameters and in particular the $\tau_W$ parameters controlling water-limited leaf area. SIF constraint is primarily propagated through the $\tau_W$ parameters onto GPP resulting in a well-constrained posterior with a $\sigma$ uncertainty of 1.6 PgC yr-1 in annual GPP of the Tropics. Although moderate constraint is seen in the key PFT-specific parameter $V_{cmax}$ for the dominant tropical PFTs (see Figure 1), in the posterior these parameters contribute to roughly 35% of the uncertainty in annual GPP.

## 3.4 Diurnal SIF Constraint

With this setup it is possible to test how the SIF-constraint on GPP might change with alternative observational times. Considering this, we test how the constraint on GPP changes when assimilating observations of SIF from alternative times of the day, assuming the same number of observations and the same observational uncertainty as used above. From this we see that different observing times yield differences in the posterior uncertainty and the effective constraint of GPP (see Figure 8). The constraint on global annual GPP when using SIF-observing times between 9:00 a.m. and 3:00 p.m. is quite similar, with the posterior uncertainty in global annual GPP ranging from $5.0 \, \mathrm{PgCyr}^{-1}$ (effective constraint of 74%) to $6.0 \, \mathrm{PgCyr}^{-1}$ (effective constraint of 68%). The most significant constraint on GPP is obtained when using SIF observations at between 11:00 or 13:00, nearest to the peak in the diurnal cycle of both GPP and SIF.

We also test the effect of utilizing SIF measurements at multiple times of the day simultaneously. We select the times 8:00 a.m., 12 noon, and 4:00 p.m., replicating a theoretical geostationary satellite. For this experiment we first test the effect of increasing the number of observations by a factor of three, assuming the same uncertainty for the three observation times.

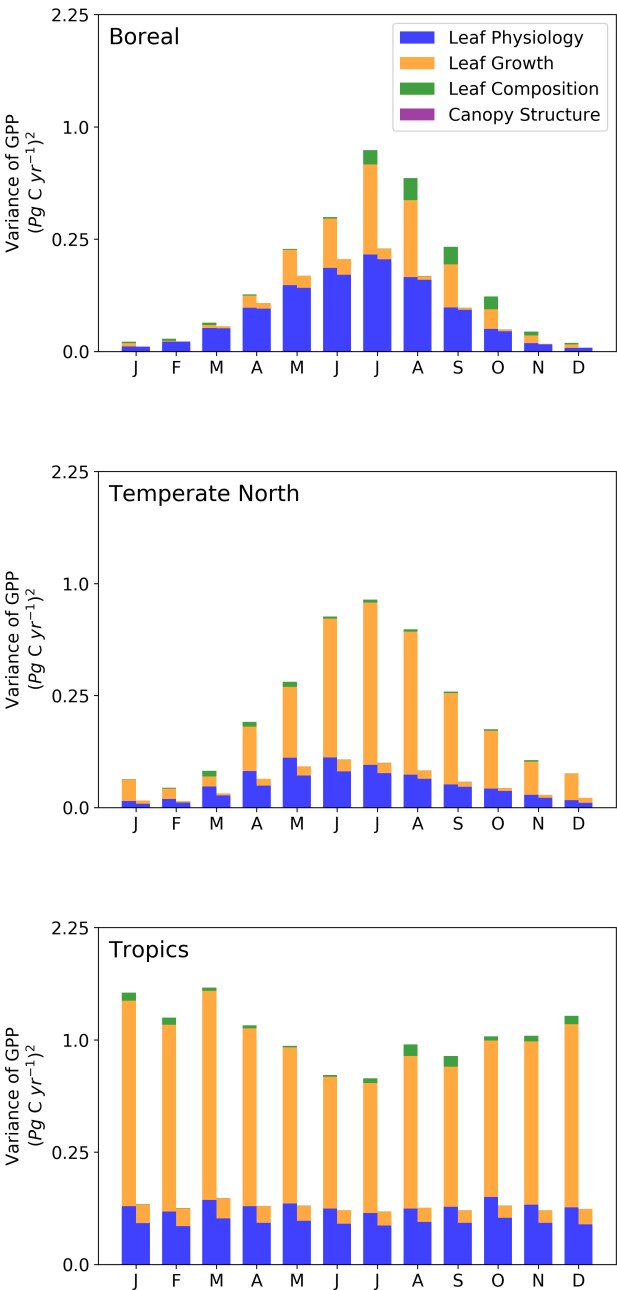

**Figure 7.** Contribution of parameter classes to parametric uncertainty in monthly GPP for three regions (see Table A1 for details on these parameter classes). For each month, the bar on the left is the prior and the bar on the right is the posterior. Uncertainties are represented as variances, thus the units are in $\mathrm{PgCyr}^{-1}$ squared and, for clarity, the y axes are on a quadratic-transformed scale.

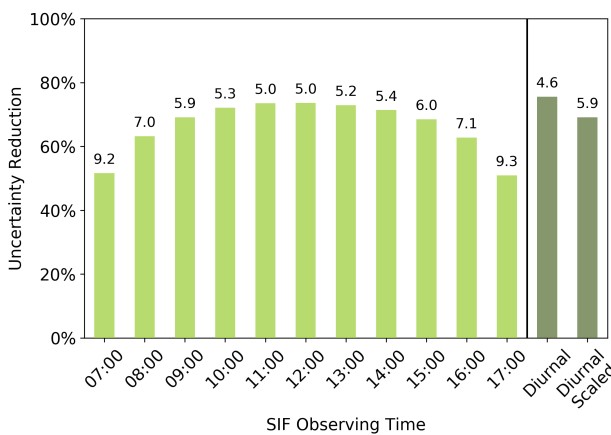

**Figure 8.** Effective constraint on global annual GPP for different observing times and the two diurnal cycle configurations. Values at the top of the bars correspond to the posterior uncertainty ($\sigma$) in global annual GPP.

Second, we also increase the number of observations by a factor of three, but scale the variance of these observations by one third. Using this second test we can assess whether differences in parameter sensitivities of SIF and GPP at the different times of the day adds value in the overall constraint.

Using a diurnal cycle of observations results in a posterior uncertainty of 4.6 $\mathrm{PgCyr}^{-1}$, or an effective constraint of 76% as in Figure 8. This is an extra 2% constraint on the uncertainty in GPP compared with observations at 12:00 noon alone. If we use a diurnal cycle of observations with scaled uncertainties, we see a slightly reduced constraint on GPP where the posterior uncertainty is 5.9 $\mathrm{PgCyr}^{-1}$ equivalent to an effective constraint of 69% (Figure 8).

### 3.5    Incorporating Uncertainty in Radiation

In order to assess the effects of incorporating uncertainty in SWRad we conduct three experiments. First is a control run, equivalent to using SIF at 1:00 p.m. as before. Second includes uncertainty in SWRad by adding it into the posterior uncertainty calculation, what might be done normally when accounting for uncertainty in forcing. Third is incorporating uncertainty in SWRad into the error propagation system with SIF, such that it's uncertainty may be constrained. This third experiment effectively treats SWRad as a model parameter by adding an extra row and column to $C_x$.

Including the uncertainty in SWRad in the calculation of posterior uncertainty in GPP results in an additional 0.03 $\mathrm{PgCyr}^{-1}$ to the prior uncertainty in global annual GPP. This is a small effect relative to the parametric uncertainties. Moreover, if we incorporate SWRad uncertainty into the error propagation system we see that this additional uncertainty is mitigated by the SIF constraint. With SWRad uncertainty included, the posterior uncertainty in GPP remains at 5.15 $\mathrm{PgCyr}^{-1}$, equivalent to the case without accounting for uncertainty in SWRad, in both cases resulting in a relative reduction of the GPP uncertainty by

72.9%. This mitigation of the additional uncertainty from SWRad is possible because both SIF and GPP are strongly sensitive to it, thus any constraint on SWRad from SIF is also propagated through to GPP.

**Table 2.** Parametric uncertainty and effective constraint for each of the SWDown experiments. Prior and posterior values shown are the one standard deviation ($\sigma$) uncertainty in global annual GPP.

| Experiment | Prior GPP ($\mathrm{PgCyr}^{-1}$) | Posterior GPP ($\mathrm{PgCyr}^{-1}$) | Effective Constraint |
|---|---|---|---|
| Control | 19.01 | 5.15 | 72.9% |
| Control+SWRad | 19.04 | 5.29 | 72.2% |
| With SIF Constraint | 19.04 | 5.15 | 72.9% |

By assessing the prior and posterior uncertainty in SWRad in $C_{x_{prior}}$ and $C_{x_{post}}$, respectively, we can assess the effective constraint following use of SIF in the error propagation system. We find that SIF constrains the SWRad uncertainty by about 29%. This gain in information on SWRad naturally results in less information being available for other parameters. The relative uncertainty reduction for most parameters decreases by a few percent. For example most $C_{ab}$ parameters see a decrease in effective constraint of around 1% and $V_{cmax}$ parameters up to 3%. With GPP exhibiting low sensitivity to $C_{ab}$ parameters and strong sensitivity to SWRad, the transfer of information from $C_{ab}$ to SWRad results in an overall mitigated effect of SWRad uncertainty on GPP.

## 4  Discussion

The results presented show that with one year of satellite SIF data observed at the GOSAT and OCO-2 satellite overpass time and SIF retrieval wavelength we can constrain a large portion of the BETHY-SCOPE parameter space and ultimately yield a parametric uncertainty in global annual GPP of $\pm 5.2\,\mathrm{PgCyr}^{-1}$. The parametric uncertainty in the prior is approximately 12% of the global annual GPP and following the addition of SIF information this is reduced to about 3% of global annual GPP. This constitutes reduction in parametric uncertainty of 73% relative to the prior. Although this data-driven constraint is model dependent, it is improved on the often-reported uncertainty of $\pm 8\,\mathrm{PgCyr}^{-1}$ from the empirical-model-based upscaled product of Beer et al. (2010).

We note that our analysis is an underestimate of the constraint, as it is performed with uncertainties calculated from the GOSAT SIF $3° \times 3°$ spatial resolution observations. With the use of higher resolution observations such as those from OCO-2 the constraint will get stronger. Similarly, with a longer time-series of data there will be stronger constraint. This occurs because the number of independent observations increases while the number of parameters remain constant.

This error propagation analysis does not assess how model SIF compares with observed SIF. However, our finding that the $\chi_r^2$ is near to the optimal value of one provides evidence that the range of possible model SIF realizations, given our

assumptions of parameter and data uncertainties, can provide coverage of the observed SIF. While this is not evidence that each specific uncertainty (e.g. parameters, model, measurement) is optimal (Michalak et al., 2005) it does suggest that overall the assumptions are valid and that we are not overconfident in, or underestimating covariances. We reiterate that the $\chi_r^2$ test

is performed using higher spatial resolution model and measured data because this is the resolution that would be applied in an assimilation of the data, and this reduces effects of representation errors. The error propagation analysis, however, benefits from using low-resolution as it greatly improves computational efficiency considering the simulations and calculations are computationally demanding.

    We also find that the effect of incorporating an error in the zero-level offset correction in the SIF observations is negligible

on posterior parametric uncertainties. This may be negligible because, for a given season, this systematic uncertainty applies across all data points, thus it scales all of the SIF values and therefore the sensitivities as well. In any case, the systematic error in the zero-level offset corrected data assessed here (Appendix Figure 10) appears small.

    The constraint on global GPP is similar when assimilating SIF at any time between 9:00 a.m. and 3:00 p.m.. Assimilating observations at the daily maximum of SIF and GPP provides the strongest constraint as both quantities exhibit the strongest

parameter sensitivities at these times. Depending upon the state of the vegetation and the environmental stress conditions, maximum SIF and GPP may occur anywhere between mid-morning and early afternoon. Therefore, we expect that effective use of different satellite-retrieved SIF observations for assimilation studies will depend not so much on their observing time but more on the spatiotemporal resolution, measurement precision, and subsequent uncertainty.

    A confounding factor to this expectation is the uncertain role of physiological stress on the diurnal cycle of SIF and GPP

and on modeling capabilities of these processes. Multiple studies have shown that various forms of environmental stress result in downregulation of PSII and changes in the fluorescence yield, particularly evident across the diurnal cycle (Carter et al., 2004; Daumard et al., 2010; Flexas et al., 1999, 2000, 2002; Freedman et al., 2002). By ingesting SIF observations at multiple times of the day we hypothesized that there could be improvements in the overall constraint on GPP as the SIF observations would capture the vegetation in different states of stress. We saw only minor improvements in the constraint and less constraint

if we assumed no additional information in the observations (i.e. with scaled uncertainty). Thus, the difference in model parameter sensitivities of SIF and GPP at other times across the diurnal cycle were not sufficient to add value to the constraint. Additionally, the constraint is worse with these scaled observational uncertainties as we are effectively removing some useful observational information at midday, the time that provides the highest sensitivities, and getting extra observational information at the lower-sensitivity times of 8:00 a.m. and 4:00 p.m.. This may be due to limitations of the model. Although BETHY-SCOPE

simulates light-induced downregulation of PSII, there is no mechanism present to simulate other forms of stress that might be expected to emerge across the diurnal cycle. However, even with a perfect model, the spatial footprint and spatiotemporal averaging of satellite observations may smooth over stress signals. Considering these confounding factors, ingesting individual SIF soundings could help remedy this problem, and there is no technical reason other than the high computational requirements that would prevent a data assimilation system from doing so.

The constraint of SIF on GPP occurs via multiple processes including leaf growth, leaf composition, physiology, and canopy structure. For the prior, uncertainty in global GPP is dominated by leaf growth processes. There is a clear and direct link

between leaf growth processes and GPP (Baldocchi, 2008) as the dynamics of leaf area influences canopy APAR which in turn strongly influences GPP. Leaf growth parameter uncertainties are relatively large in the prior, with coefficients of variation up to 50%. It is perhaps no surprise then that these parameters project a large uncertainty onto GPP. Regardless, both GPP and SIF respond similarly to the leaf growth parameters so information from observations of SIF can provide direct constraint on GPP in this way. Many leaf growth parameters, particularly for grasses, crops, and deciduous trees and shrubs, receive constraint of >40% from SIF thus the overall contribution of leaf growth parameters in the posterior is considerably reduced.

Of particular importance is the parameter describing water limitation on leaf growth ($\tau_W$), which accounts for about 80% of the prior uncertainty in global GPP. Model SIF and GPP are highly sensitive to this parameter hence there are large values in $H$ and $H_{GPP}$ pertaining to $\tau_W$. This relates to the model formulation as many of the leaf growth parameters determine phenological processes such as temperature or light dependent growth triggers (i.e. temporal evolution of leaf area), while $\tau_W$ is the only process parameter controlling leaf area other than intrinsic maximum LAI ($\tilde{\Lambda}$) (Knorr et al., 2010). Additionally, as we assume little prior knowledge for $\tau_W$ (i.e. it is highly uncertain) it projects a relatively large uncertainty onto GPP.

At the global scale, $\tau_W$ for crops, C3 grasses and C4 grasses ($\tau_W^{Gr}$) is particularly important. Combined, these three PFTs cover about 47% of the land surface and account for just over 50% of global annual GPP in the present model setup. Although this contribution to global GPP may seem high, it is based on the prior estimate. In a recent study by Scholze et al. (2016) where atmospheric $CO_2$ concentration and SMOS soil moisture were assimilated into BETHY, the posterior value for $\tau_W^{Gr}$ shifted approximately three standard deviations away from the prior, the result of which would have been a large change in the GPP of these PFTs. This exposes a limitation to the present study as we can predict and quantify how SIF will constrain the uncertainty of process parameters and GPP, but we cannot predict how their values will change.

The constraint SIF provides on leaf growth processes is also perhaps achievable from other remote sensing products such as FAPAR (e.g. Kaminski et al., 2012). A direct comparative study would be required to assess the advantages and disadvantages of each observational constraint. Nevertheless, issues arise with these alternative observations when observing dense canopies (Yang et al., 2015) or vegetation with high photosynthetic rates such as crops as they are near saturation (Guanter et al., 2014). Information on maximum potential LAI ($\tilde{\Lambda}$) and parameters pertaining to understorey shrubs and grasses are therefore also limited (Knorr et al., 2010). A strong benefit of SIF is that it shows minimal saturation effects (e.g. Yang et al., 2015), especially beyond 700 nm where most current satellite SIF measurements are made.

The strong constraint SIF provides on leaf growth processes indicates that it is likely to provide improved monitoring of key phenological processes such as the timing of leaf onset, leaf sensescence and growing season length. This will be highly useful in interpreting results from a full assimilation with SIF as the posterior process parameter values can be compared with independent ecophysiological data, taking consideration of spatial scale issues.

Beyond observing LAI dynamics SIF can also provide critical insights into physiological processes (e.g. Walther et al., 2016). We see here that SIF provides weak to moderate constraint on a range of physiological parameters, including up to 30% constraint on $V_{cmax}$ parameters. The limited constraint on these parameters results in the posterior being dominated by uncertainty in the parameters representing physiological processes. This is in line with Koffi et al. (2015) who found limited sensitivity of simulated SIF to $V_{cmax}$. We note that under certain conditions, where other key variables are well known, SIF

can be used to retrieve $V_{cmax}$ (Zhang et al., 2014). The ability of SIF to inform on physiological processes at all will provide researchers with a powerful new insight into spatiotemporal patterns of GPP. As was shown by Walther et al. (2016) and Yang et al. (2015) this is particularly important for evergreen vegetation as changes in photosynthetic activity are not always reflected

by changes in traditional vegetation indices.

Chlorophyll content here constitutes a classic nuisance variable. A nuisance variable is one that is not perfectly known, impacts the observations we wish to use but not the target variable (Rayner et al., 2005). However, exploiting the well-documented correlation between leaf nitrogen content, $V_{cmax}$, and $C_{ab}$ may help curtail this problem (Evans, 1989; Kattge et al., 2009). Houborg et al. (2013) demonstrated that by including a semi-mechanistic relationship between these variables in the Com-

munity Land Model and using satellite-based estimates of chlorophyll to derive $V_{cmax}$, there is significant improvement in predictions of carbon fluxes over a field site. Implementing such a semi-mechanistic link in a data assimilation system would enable the strong constraint that SIF provides on $C_{ab}$ to feed more directly onto GPP. However, in this study it is assumed $C_{ab}$ and $V_{cmax}$ can be resolved independently which may not be the case considering ecophysiological studies have shown the two parameters are commonly correlated.

Almost all terrestrial carbon cycle models use down-welling radiation at the Earth's surface as an input variable. Any uncertainty in this forcing will translate into uncertainty in carbon fluxes including GPP, and few studies consider such uncertainties. A known systematic error (i.e. bias) in forcing variables (e.g. Boilley and Wald, 2015) cannot be considered in the present error propagation system, however, in such a case a correction to the data should be performed as it will bias carbon flux estimates. For random errors that cannot be removed however, they may be considered in the uncertainty of carbon flux estimates using

error propagation. At the global scale, Kato et al. (2012) used a perturbation study, along with modeled irradiance and remotely sensed measurements to compute a random error ($\sigma$) of 12 $Wm^{-2}$ for monthly gridded downward shortwave radiation over the land. We considered this uncertainty by incorporating it into the error propagation system with SIF. While including this forcing uncertainty in the prior increases the prior uncertainty of GPP, incorporating the former into the error propagation analysis with the SIF observations mitigates the downstream effect on GPP. SIF can therefore provide useful information on

the SWRad forcing via a data assimilation system. The consideration of uncertainties in forcing variables such as SWRad on terrestrial carbon fluxes is important when estimating the uncertainty in GPP. However, the effect on uncertainty in GPP may be strongly reduced by using SIF observations.

The results presented here demonstrate how SIF observations may be utilized to optimize a process-based terrestrial biosphere model and constrain uncertainty of simulated GPP. These results are, however, model dependent. The assumption is that

the model simulates the most important processes driving SIF and GPP. Some key, remaining unknowns include how processes such as environmental stress, 3-dimensional canopy structure effects, or nitrogen cycling may affect the SIF signal. As better understanding is developed on the role that these processes play, modeling capabilities will also be improved. Additionally, a different set of prior parameter values will alter the results due to changes in the Jacobian. Use of prior knowledge, based on ecophysiological data and its probable range, is critical to curtail this problem. The choice of how to spatially differentiate

the parameters will also affect results (Ziehn et al., 2011). Selecting an optimal parameter set that has the fewest degrees of freedom, yet provides the best fit to the observational data is outside the scope of this study however. Implementation of a

parameter estimation scheme in a full data assimilation system with SIF and other observational data will help address these challenges. Earlier work by Koffi et al. (2015) demonstrated that the model can simulate the patterns of observed satellite SIF quite well, indicating the model can ingest the data. Further work will be needed to assess how well the model can simulate patterns of SIF with an optimized, realistic parameter set.

## 5 Conclusions

We assessed the ability of satellite SIF observations to constrain uncertainty in model parameters and uncertainty in spatiotemporal patterns of simulated GPP using a process-based terrestrial biosphere model. The results show that there is strong constraint of parametric uncertainties across a wide range of processes including leaf growth dynamics and leaf physiology when assimilating just one year of SIF observations. Combined, the SIF constraint on parametric uncertainties propagates through to a strong reduction of uncertainty in GPP. The prior uncertainty in global annual GPP is reduced by 73% from 19.0 $PgCyr^{-1}$ to 5.2 $PgCyr^{-1}$. Although model dependent, this result demonstrates the potential of SIF observations to improve our understanding of GPP. We also showed that a data assimilation framework with error propagation such as this allows us to account for uncertainty in model forcing such as SWRad. Surprisingly, by including it into this framework with SIF observations there is a net-zero effect on uncertainty in GPP due to the sensitivity of both SIF and GPP to radiation. This study is a crucial first step toward assimilating satellite SIF data to estimate spatiotemporal patterns of GPP. With the addition of other observational constraints such as atmospheric $CO_2$ concentration or soil moisture there is also the possibility of accurately disaggregating the net carbon flux into its component fluxes, GPP and ecosystem respiration. Indeed, with these additional, complementary observations of the terrestrial biosphere further constraint could be gained as other regions of parameter space can be resolved (Scholze et al., 2016).

## 6 Code availability

The BETHY-SCOPE model code is available in the repository at https://github.com/NortonAlex/BETHY-SCOPE-Interactive-Phenology. The GOSAT satellite SIF data used in this paper is from the ACOS project (version b35).

## Appendix A

## A1 Model Process Parameters

**Table A1.** BETHY-SCOPE process parameters along with their prior and optimized uncertainties following SIF constraint, represented as one standard deviation. Relative uncertainty reduction (i.e. effective constraint) is reported for the error propagation with low-resolution and high-resolution SIF observations. Units are: $V_{cmax}$, μmol$(CO_2)$ m$^{-2}$ s$^{-1}$; $a_{V_o,V_c}$ and $a_{R_d,V_c}$, dimensionless ratios; $K_C$ and $K_O$, bar; $\tilde{\Lambda}$, m$^2$ m$^{-2}$; $T_\phi$, °C; $T_r$, °C; $t_c$, hours; $t_r$, hours; $\xi$, d$^{-1}$; $k_L$, d$^{-1}$; $\tau_W$, days; $C_{ab}$, μg cm$^{-2}$; $C_{dm}$, g cm$^{-2}$; $C_{sm}$, dimensionless fraction; hc, m; leaf width, m.

| Class | # | Description | Parameter | Prior Mean | Prior Uncertainty | Effective Constraint (%) |
|---|---|---|---|---|---|---|
| LEAF PHYSIOLOGY | 1 | | $V_{cmax}$ (TrEv) | 60 | 12 | 19.4 |
| | 2 | | $V_{cmax}$ (TrDec) | 90 | 18 | 12.4 |
| | 3 | | $V_{cmax}$ (TmpEv) | 41 | 8.2 | 0.3 |
| | 4 | | $V_{cmax}$ (TmpDec) | 35 | 7 | <0.1 |
| | 5 | | $V_{cmax}$ (EvCn) | 29 | 5.8 | 0.3 |
| | 6 | Maximum | $V_{cmax}$ (DecCn) | 53 | 10.6 | <0.1 |
| | 7 | carboxylation rate | $V_{cmax}$ (EvShr) | 52 | 10.4 | 21.3 |
| | 8 | at 25°C | $V_{cmax}$ (DecShr) | 160 | 32 | 0.2 |
| | 9 | | $V_{cmax}$ (C3Gr) | 42 | 8.4 | 30.7 |
| | 10 | | $V_{cmax}$ (C4Gr) | 8 | 1.6 | 28.5 |
| | 11 | | $V_{cmax}$ (Tund) | 20 | 4 | 0.5 |
| | 12 | | $V_{cmax}$ (Wetl) | 20 | 4 | <0.1 |
| | 13 | | $V_{cmax}$ (Crop) | 117 | 23.4 | 6.0 |
| | 14 | Ratio of $V_{omax}$ to $V_{cmax}$ | $a_{V_o,V_c}$ | 0.22 | 0.0022 | <0.1 |
| | 15 | Ratio of $R_d$ to $V_{cmax}$ | $a_{R_d,V_c}$ | 0.015 | 0.0015 | <0.1 |
| | 16 | Michaelis-Menten constant of Rubisco for $CO_2$ | $K_C$ | 350e-6 | 23e-6 | 0.9 |
| | 17 | Michaelis-Menten constant of Rubisco for $O_2$ | $K_O$ | 0.45 | 0.0165 | <0.1 |
| LEAF GROWTH | 18 | Max. leaf area index | $\tilde{\Lambda}$ | 5 | 0.25 | 7.0 |
| | 19 | | $T_\phi$ (4) | 10 | 0.5 | 8.9 |
| | 20 | | $T_\phi$ (5,6,11) | 10 | 0.5 | 16.6 |
| | 21 | Temperature at leaf onset | $T_\phi$ (8) | 8 | 0.5 | 2.3 |
| | 22 | | $T_\phi$ (9,10,12) | 2 | 0.5 | 46.8 |
| | 23 | | $T_\phi$ (13) | 15 | 1 | 49.6 |

| | # | Description | Parameter | | | |
|---|---|---|---|---|---|---|
| **LEAF GROWTH** | 24 | | $T_r$ (4,8,13) | 2 | 0.1 | 1.9 |
| | 25 | Spatial range ($1\sigma$) of $T_\phi$ | $T_r$ (5,6,11) | 2 | 0.1 | 0.8 |
| | 26 | | $T_r$ (9,10,12) | 0.5 | 0.1 | 9.6 |
| | 27 | Day length at leaf shedding | $t_c$ (4-6,8,11) | 10.5 | 0.5 | 32.1 |
| | 28 | Spatial range ($1\sigma$) of $t_c$ | $t_r$ (4-6,8,11) | 0.5 | 0.1 | 1.4 |
| | 29 | Initial linear leaf growth | $\xi$ | 0.5 | 0.1 | 20.0 |
| | 30 | Inverse of leaf longevity | $k_L$ (2,4,6,8,9,10,12,13) | 0.1 | 0.05 | 58.5 |
| | 31 | | $k_L$ (5,11) | 3e-3 | 0.5e-3 | 11.2 |
| | 32 | Length of dry spell before leaf shedding | $\tau_W$ (1,3,7) | 180 | 60 | 56.9 |
| | 33 | | $\tau_W$ (2) | 90 | 30 | 37.9 |
| | 34 | | $\tau_W$ (9,10,12,13) | 30 | 15 | 64.6 |
| **LEAF COMPOSITION** | 35 | | $C_{ab}$ (TrEv) | 40 | 20 | 7.4 |
| | 36 | | $C_{ab}$ (TrDec) | 15 | 20 | 60.7 |
| | 37 | | $C_{ab}$ (TmpEv) | 15 | 20 | 31.3 |
| | 38 | | $C_{ab}$ (TmpDec) | 10 | 20 | 69.7 |
| | 39 | | $C_{ab}$ (EvCn) | 10 | 20 | 73.8 |
| | 40 | | $C_{ab}$ (DecCn) | 10 | 20 | 66.9 |
| | 41 | Chlorophyll $ab$ content | $C_{ab}$ (EvShr) | 10 | 20 | 72.5 |
| | 42 | | $C_{ab}$ (DecShr) | 10 | 20 | 66.3 |
| | 43 | | $C_{ab}$ (C3Gr) | 10 | 20 | 73.9 |
| | 44 | | $C_{ab}$ (C4Gr) | 5 | 20 | 83.6 |
| | 45 | | $C_{ab}$ (Tund) | 10 | 20 | 72.9 |
| | 46 | | $C_{ab}$ (Wetl) | 10 | 20 | 49.6 |
| | 47 | | $C_{ab}$ (Crop) | 20 | 20 | 54.6 |
| | 48 | Dry matter content | $C_{dm}$ | 0.012 | 0.002 | 1.3 |
| | 49 | Senescent material content | $C_{sm}$ | 0 | 0.01 | 0.2 |
| **CANOPY STRUCTURE** | 50 | Leaf inclination distribution | $LIDFa$ | -0.35 | 0.1 | 21.5 |
| | 51 | function parameters | $LIDFb$ | -0.15 | 0.1 | 9.0 |
| | 52 | Vegetation height | hc | 1 | 0.5 | 6.8 |
| | 53 | | leaf width | 0.1 | 0.01 | 0.3 |

## A2 GOSAT SIF Uncertainty Calculations

To get the variance of a target grid cell at the model grid resolution (ylat,xlon) we first determine the area-weighted variance of each GOSAT grid cell (ilat,jlon) within that target grid cell. The area-weighting per GOSAT grid cell ($\hat{Area}_{ilat,jlon}$) is calculated as the area divided by the total area of the target grid cell. This enables us to account for different grid cell sizes

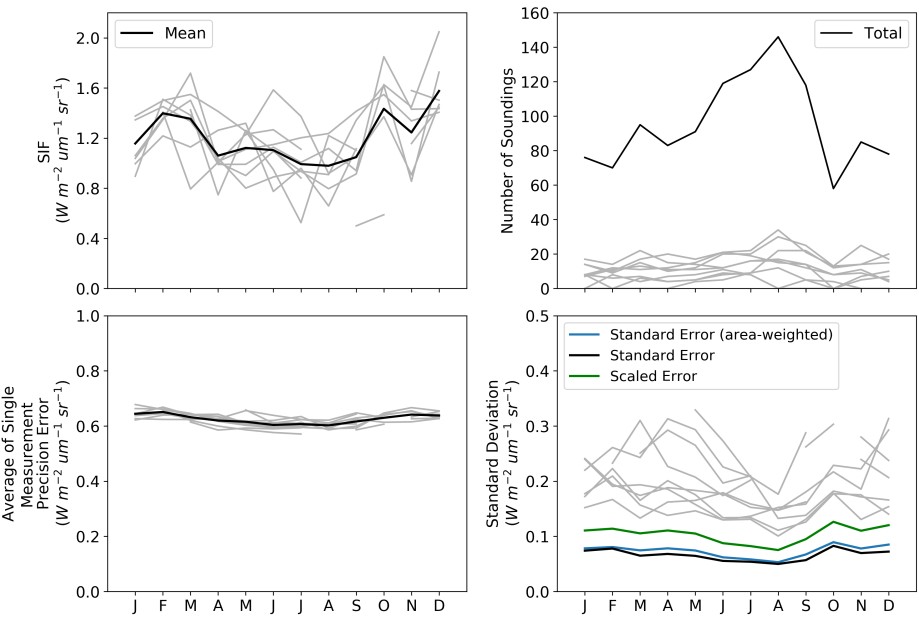

**Figure 9.** An example of the GOSAT SIF data and uncertainty calculations over a low-resolution model grid cell centered over the Amazon forest at 3.75°S and 65°W. Grey lines show individual 3° × 3°GOSAT grid cells. Black lines show the aggregated data for the 7.5° × 10°model grid cell. Bottom shows the calculated uncertainty (standard deviation) at the model grid resolution in black, blue and green. The black line is the standard error calculated using Eq. 5; the blue line is the standard error calculated using Eq. A1; the green line is the same as the blue but scaled by $\sqrt{2}$ to account for correlated errors which is used in this study.

considering SIF is in physical units per unit area. We then sum the area-weighted variances and scale this uncertainty by the square root of two (see equation 5). Scaling the uncertainty in this way effectively doubles the variance in an independent dimension.

$$\sigma^2_{ylat,xlon} = \sqrt{2} \sum (A\hat{r}ea^2_{ilat,jlon} \cdot \sigma^2_{ilat,jlon}) \tag{A1}$$

## A3   Systematic Error in GOSAT SIF Observations

*Acknowledgements.* A. Norton was partly supported by an Australian Postgraduate Award provided by the Australian Government and a CSIRO OCE Scholarship. The research was funded, in part, by the ARC Center of Excellence for Climate System Science (grant CE110001028).

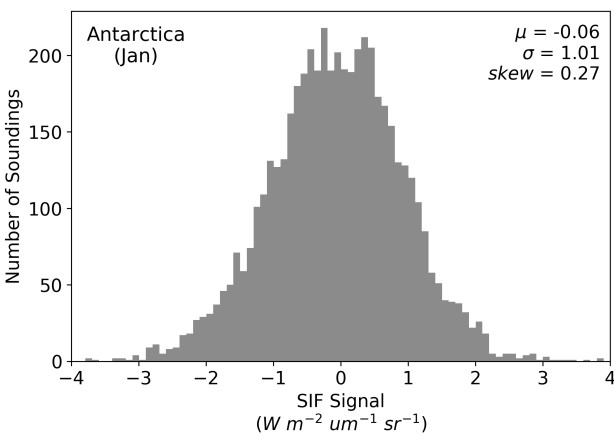

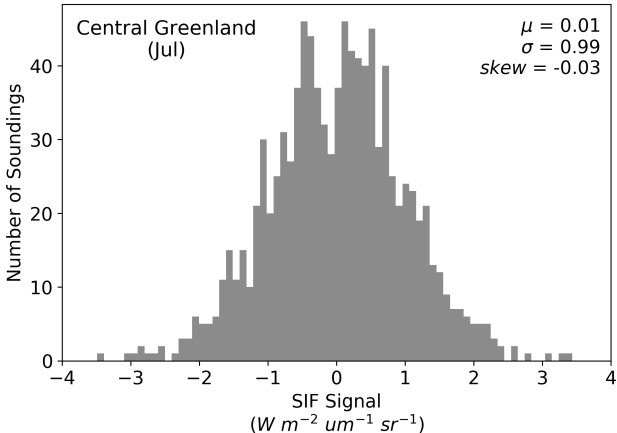

**Figure 10.** Analysis of systematic errors in the GOSAT SIF observations. We assess the zero-level offset corrected GOSAT SIF soundings over two ice-covered and therefore non-fluorescent regions. The first is Antarctica in January, between latitudes $70°$S to $80°$S and longitudes $75°$W to $155°$E. The second is central Greenland in July, between latitudes $73°$N to $80°$N and longitudes $30°$W to $52°$W. With no systematic error the mean ($\mu$) value of the distribution should be on zero. As is shown, $\mu$ is non-zero and varies in sign and magnitude between January and July. This test samples the error distribution in the zero-level offset (i.e. $\varepsilon_z$ in Eq. 6).

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
