# Peer review of "Assimilating solar-induced chlorophyll fluorescence into the terrestrial biosphere model BETHY-SCOPE v1.0: Model description and information content"

_Geoscientific Model Development, 2017_

## Short Comment (SC1) · 24 Feb 2017

Dear authors,

in my role as Executive editor of GMD, I would like to bring to your attention our Editorial version 1.1:

http://www.geosci-model-dev.net/8/3487/2015/gmd-8-3487-2015.html

This highlights some requirements of papers published in GMD, which is also available on the GMD website in the 'Manuscript Types' section:

http://www.geoscientific-model-development.net/submission/manuscript_types.html

In particular, please note that for your paper, the following requirement has not been met in the Discussions paper:

- "The main paper must give the model name and version number (or other unique identifier) in the title."

Please add a version number for BETHY-SCOPE in the title upon your revised submission to GMD.

Yours,

Astrid Kerkweg

---

## Referee Comment (RC1) · Anonymous Referee #1 · 21 Apr 2017

Comment: Assimilating solar-induced chlorophyll fluorescence into the terrestrial biosphere model BETHY-SCOPE: Model description and information content

Authors: Alexander Norton et al.

Summary: This paper uses satellite observations of Solar-Induced Fluorescence (SIF) in an inversion scheme (CCDAS) to reduce uncertainty in *a posteriori* estimates of model parameters and outputs, specifically GPP. Interestingly, no attention is given to actual parameter values or GPP estimates; the focus is entirely on how much reduction in uncertainty can be expected due to the inclusion of SIF.

The paper is reasonably well written, and uses a novel approach to attempt to reduce uncertainty in *a posteriori* estimates of model parameters and output. However, I feel that the paper needs clarification and perhaps some reorganization to help readers to follow the story. Furthermore, I believe that the critical issue of observational uncertainty is given too little attention and must be clarified.

The authors provide reasonably comprehensive citations for CCDAS, but the paper reads is if it were written (as it probably was) by someone who is a Data Assimilation (DA) expert. To this reviewer it seems that some details are either implied or 'skipped over'. It is likely that many readers will be DA experts themselves, but the inclusion of SIF will probably draw in readership that may not possess the DA expertise to easily understand what is going on. I may be a member of that part of the audience, so some clarification is warranted. Specifically, the relationship between covariance matrices ($C_x$, $C_d$) and standard deviation ($\sigma$) is not entirely clear.

The description of grids used and observation area ("GOSAT grid cell"; section 2.4) needs clarification. Two grid sizes are mentioned in Section 2.4, but we don't learn much more about them until Section 2.5. I would like to see a more deliberate explanation of "here is what we are going to do, and here is how we are going to do it". That might fit better in Section 2.1. Some specific Issues:

- Figure 3, showing observational uncertainty, is not referred to in the section describing observational uncertainty. It needs to be.

**Observational uncertainty**, Eqn 4: I see two ways that this value can be small: 1) there are many observations, and $\sigma^2$ is small. 2) There are very few observations, and Area is small. Parazoo et al. (2013) estimated uncertainty as the standard error. This has the effect of allowing a large error in regions with very few observations, like the tropics. Figure 3 in the manuscript under review shows some of the smallest observational uncertainty in the tropics, and that makes absolutely no sense to me. I've worked with the GOSAT data, and over the deepest tropics there are very few observations, which makes me suspicious that your uncertainty is small because of reason 2). Parazoo et al. did not extend their analysis to the wetter parts of Amazonia because they just didn't have enough data to justify it. Now the authors claim that this region has some of the smallest observational uncertainty on the

globe! A detailed justification of how uncertainty can be very small over a region with few or no datapoints is an absolute necessity.

I do not think multiplying by square-root-2 is sufficient to remedy what might be unrealistically low uncertainty values.

When GOSAT 2010 data is aggregated onto the 1.25x1.0 degree MERRA grid, I see that the maximum number of retrievals for a given month, anywhere on the globe, is between 30-35 or so. Looking at South America, I see that very few MERRA gridcells have more than 10 retrievals in a given month during 2010, and many gridcells have 5 or fewer. Aggregating up to 7x10 (or 2x2) you are not going to get very much increase in sample size. I'd like to see the authors address the sparseness of the GOSAT data and explain how this will or will not effect their method.

The number of GOSAT observations is invariant and does not change with grid size. The aggregation of GOSAT observations changes with grid size (Section 2.4). This should be clarified.

An individual GOSAT retrieval has pixel size of around 10 km$^2$, I believe. OCO-2 will have a pixel size of ~5 km$^2$, and GOME-2 is a 40-80 pixel, or 3200 km$^2$. This will have a large impact on your inversion scheme and the calculation of observational uncertainty. Since this paper only uses GOSAT, the other products probably don't need too much (or any?) explanation, but I do have questions about GOSAT and the grids used:

1. There is the possibility for (possibly) many 10km$^2$ GOSAT retrievals to be included in a 7.5x10 degree gridcell. For that matter there can be many of them in a 2x2 gridcell too. BETHY-SCOPE tiles in 3 PFTs; how are GOSAT retrievals registered to these PFTs? Are GOSAT retrievals marked with a specific land cover type, and accumulated on a per-PFT basis? What about GOSAT retrievals that are not associated with one of the 3 PFTs tiled into the BETHY-SCOPE gridcell? Are they discarded? Why or why not?
2. If all GOSAT retrievals within a gridcell are utilized, is the mean taken and used for DA with all 3 PFTs? In this case aren't you 'smearing out' the information that SIF provides? Guanter et al. (2012) demonstrate that the linear relationship between SIF and individual PFTs is heterogeneous. Do you take this into account? If so, how? If not, why not?
3. In August 2010 the GOSAT scan strategy was changed; the area observed was decreased, but the number of retrievals over a given region was increased. How does this effect the two questions above?

The reduction in uncertainty for global GPP is dramatic (79%). However, this reduction is critically dependent upon $C_d$ (observation uncertainty) according to equation 1. Therefore, I think it is absolutely essential that the questions

surrounding the determination of this observation uncertainty are answered in a clear and categorical manner.

I'm not a DA expert, but I do collaborate with quite a few people who are, and I think I understand the basics. The covariance matrices are absolutely fundamental to the outcomes of a DA experiment: If the observational uncertainty is small and the model uncertainty large, the *a posteriori* outcome can be pulled strongly towards the observations. If the opposite is true, then it will be hard to budge the inversion away from the model prior. Is this correct?

In this paper the first case is presented: the observational uncertainty is, to my eye extremely small and therefore results in an amazing reduction in uncertainty in the *a posteriori* result.

The absence of evaluation of actual posterior values of either parameter or flux values may actually hinder the analysis. If the result of the study is an outlandish value for global GPP, then that might indicate a problem. Of course, estimates of global GPP vary by about a factor of two (Huntzinger et al., 2012), so maybe this wouldn't help as much as one might hope. However, posterior parameter and flux values might offer insight, and a comprehensive evaluation of method and results (values of parameters and flux) could provide more support for the authors' conclusions. Was this considered? Why or why not? I'm suspicious that posterior flux and parameter values *were* outlandish, and a choice was made to focus on method even though results may be untrustworthy. I suspect many readers will have this suspicion too.

A detailed description of the construction of the observation uncertainty may detract from the paper's readability, but including it in an appendix would be appropriate. Additionally, I would like to see, perhaps in supplemental material, a step-by-step description of the calculation of the observation uncertainty, perhaps in the 7x10 gridcell that contains Manaus, Brazil.

To see such a large reduction in error sent warning bells ringing with me; I don't think it is an overstatement to say that the entire paper depends on the observation uncertainty. If the authors can demonstrate that the values shown in Figure 3 are justifiable, then the paper has merit. If not, I think the whole endeavor falls apart, as the structural underpinning would have disintegrated. In that case the paper is not worthy of publication.

Specific Comments:
- Figure 2: The information here is too dense (small labels, tiny resolution on the plot) to follow. If the only pertinent information is in the lower-right-hand of the plot, why not omit the rest and enlarge this sector of the graph?

- Figure 2: There is very little description of the graph and what it means. Again, this may be another case where the authors are assuming that their readers look at graphs like this every day and know what it is showing.
- Figure 3: What are the units?
- Figure 4: Absolute uncertainty annual GPP will of course correlate directly with productivity. If you standardize the time series and look at relative uncertainty I imagine that map will look very different. Have you done this? If you have, do Figures 4-6 look similar or different?
- Table 1A: There is no description of what these parameters are and what they do. There are sporadic mentions in the text, but for the most part the reader is left to one's self to figure out what these parameters are for. I would like to see a column added (there appears to be room, as the uncertainty reduction columns could be re-formatted) with a couple of words or a phrase describing each variable. Section 3.2: line 14 on page 11 mentions that $\tau_W$ makes up 82% of the global annual uncertainty in posterior global GPP. The reader does not know what $\tau_W$ is. At the end of Section 3.1 there are several other parameters listed, and again the reader is not told what they are. It might be helpful to have a short description in parentheses following the listing of each parameter, but I would prefer to see that information in table 1A.
- Boilley and Wald (2015) discuss a high bias in the radiation from reanalyses. I'm not sure this is the same as the uncertainty mentioned in sections 2.4 and 3.4. Can you elaborate?
- Page 17, lines 7-8: "…we can predict and quantify how SIF wil constrain the uncertainty of process parameters and GPP, but we cannot predict how their values will change". Why not? Can't you back the posterior values out of the *a posteriori* covariance matrices and the Jacobian? Isn't the whole point of DA to obtain these posterior values?

References

Boilly and Wald, 2015: Comparison between meteorological re-analyses from ERA-Interim and MERRA and measurements of daily solar irradiation at surface. Renewable Energy, 75, 135-143.

Guanter et al 2012: Retrieval and global assessment of terrestrial chlorophyll fluorescence from GOSAT space measurements. Remote Sensing of Environment, 121, 236-251.

Parazoo, N.C., K. Bowman, C. Frankenberg, J.-E. Lee, J.B. Fisher, J. Worden, D.B.A. Jones, J. Berry, G.J. Collatz, I.T. Baker, M. Jung, J. Liu, G. Osterman, C. O'Dell, A. Butz, S. Guerlet, Y. Yoshida, H. Chen, C. Gerbig, 2013: Interpreting seasonal changes in the carbon balance of southern Amazonia using measurements of XCO2 and chlorophyll fluorescence from GOSAT. Geophys. Res. Let., 40, 2829-2833, doi:10.1002/grl.50452.

---

## Short Comment (SC3) · 12 May 2017

First, the paper is mostly well written and quite timely. Thank you for this valuable contribution.

I have two large concerns/recommendations and a few small ones.

First, the conclusions of the paper are shocking to many people familiar with modeling SIF and using it in the context explored here, to constrain GPP. The expectations of uncertainties on the order of 2.8PgC /year on GPP seem far too optimistic. Why do they come up so optimistic? I am left to believe that the assumption of a perfect model

structure and the errors only arising from the uncertainty in parameters lead to such results. For example, most models can not come close to reproducing the magnitude of GPP from SiF, only temporal dynamics. The authors' conclusions are fine given that the readers are working from the correct set of assumptions. I felt that this assumption of perfect model structure should have been introduced as a stronger caveat to warn those "abstract surfers" who rarely have time to digest an entire paper, about the practical limits of the work, as our models now stand.

Second, I won't imply there are errors in Section 2.4 but it is written in such a confusing manner that my guess is over half, if not most readers, will not be able to follow it. I don't believe there are any complicated statistics in there, but the relationships between the number of independent samples and grid observation resolution is really unclear. I recommend a complete rewrite of this section (Page 7, lines 9-26 mainly) and/or a short appendix/supp material section illustrating EXACTLY what you are talking about with a concrete example.

Pg 5/Line 29: I assume this assumption "This means that we optimize . . . quantities." is tantamount to ignoring any model error that would stand in the way of a "true" estimate of GPP from satellite SIF? If so, it should probably mentioned.

Pg 7/Line 13: Are the actual units for SIF ever mentioned?

Pg 7/Line 9: It is often hard to interpret whether the random variable of interest is the spatial variability of grid cell means or the variability of a single grid cell mean. Again, more precise terminology and definitions would often help.

Pg 7/Line 24: Again, the main problem here is that readers are used to seeing satellite observations whose associated errors are large at the single sounding level but get smaller as many samples are averaged together (larger spatial scale mean value). This text runs counter to that thinking. One is essentially *assuming* far stronger constraints on the data as you move to finer scales. Again, an example along w/ the equations would make this much more clear to all readers, let alone those w/o an extensive

statistics background.

Pg 8/Line 2: Traditionally, the issue w/ uncertainty in SW radiation has been w/ an overestimate of SW from GCM reanalysis. This is thought to result from a lack of characterization of fine scale clouds due to poor model resolution. See http://nacp.ornl.gov/docs/AGU_Ricciuto2009.pdf, not sure if Ricciuto ever published it but it's a reasonably well known problem. So, I guess the question is, how would an unknown overestimate of 20% in shortwave radiation affect the conclusions?

―――――――――――――――――

---

## Referee Comment (RC2) · Anonymous Referee #2 · 7 Jun 2017

This study evaluates the benefit of assimilating satellite-retrieved chlorophyll fluorescence into a mechanistic land surface model, to reduce the uncertainty in model parameters and simulated gross primary production (GPP). This study indeed tackles a critical issue in the current efforts towards making the most of diverse data infromation content when building efficient carbon cycle data assimilation systems.

There are, however, a few important issues in this manuscript, some of them critical. They are listed in the general comments below, followed by specific remarks/corrections.

[Figure]

**General comments**

First, while the manuscript is often fairly written, on numerous occasions sentences
are redundant, strangely formulated, thus logical progression of arguments is hard to
follow. Frankly, it sometimes feels as if the authors did not read themselves again before
submitting the manuscript. It could be just be a matter of style, but in some occasions
it simply results in a lack of clarity. While I tried to list specific parts in the *Specific
comments* and *Technical comments* section, I suggest a strong effort of rewriting in
general. That will also make the manuscript much more accessible to modellers/data
experts outside the field of CCDAS or even data assimilation at large.

Second, and perhaps more importantly, the way the observation uncertainty used in
Eq. 1 is defined is quite vague. Judging from the elements presented in Sect. 2.4,
it seems that only the 'measurements' uncertainty of GOSAT retrievals of SIF is ac-
counted for in $C_D$, neglecting the structural uncertainty ($C_T$, using the notation of Taran-
tola (1987)) of the BETHY-SCOPE model. If structural uncertainty is considered, that
should be detailed in Sect. 2.4. If $C_T$ is not taken into account, this would bear impor-
tant consequences. While $C_T$ is hard to estimate explicitly (although some diagnostic
methods exist, e.g. see Desroziers et al. (2005), applied to land surface models by
Kuppel et al. (2013)), its magnitude and structure might be comensurate or even domi-
nant over measurement uncertainties when building $C_D$. Not including it in Eq. 1 would
then largely underestimate the posterior uncertainty of parameters and, by propagation
that of modelled GPP. As noted for another reviewer, this would constitutes a serious
theoretical flaw in the scope of this study and make it unsuitable for publication.

**Specific comments**

**P2, L11-12:** This sentence is rather vague, can the authors be more precise and add

references to support this assertion?

**P2, L27-28:** Data assimilation is not only used with mechanistic models nor for terrestrial carbon cycle modeling. I suggest to reformulated, for example: "In the case of mechanistic models, this is done by constraining the simulated underlying processes.".

**P2, L28-32:** In this review of the state of the art, efforts from other groups to build "mechanistic" CCDAS might deserve to be cited as well, e.g. (Peylin et al., 2016) and the discussion/review by MacBean et al. (2016).

**P3, L4-6:** Some references would be necessary to back these assertions.

**P5, L8:** The last sentence of this paragraph feels rather clumsy, it should reformulated.

**P5, L9:** Table A1 is rather long and that is fair game given the number of parameters, yet to make it more reader-friendly I would suggest to:

- include a description column for each type of parameter,

- add the corresponding PFT between brackets for all PFT-dependent parameters, as is done for $Vc_{max}$,

- add "subsection rows" with parameter categories (leaf growth, ecophysiology etc.).

**P6, L2-3:** It is because the PDFs of parameters and observations is treated as Gaussian that it can be described by their first two moments, mean and standard deviation (taken here as the metric of uncertainty, that might need to be specified here already well), not the other way around.

**P6, L1-4:** The definition of observations here should be more precise; the reader (especially if not familiar with the data assimilation vocabulary) would assume it relates to *measured* observations (as the previous paragraph uses "SIF observations" to designate measurements), while in a rigorous probabilistic framework it should refer to quantities in the observation space (including measurements and model outputs, see *General comments*).

**P6, L12-13:** I guess that the authors meant with this sentence that a) in a linear world $H$ is independent from $x$, but b) this is an oversimplication, therefore c) bringing limitation in accuracy to a method relying on $H(x_0)$ to approximate $H(x_{post})$. It is not clear at all from the current formulation, which even almost suggest that because of linearity the choice of $x_0$ can influence the results (through a changing $H$)...

**P6, L13-21:** I am not sure how "the use of prior knowledge limits the effect of this problem": is it because we assume that the posterior parameters values will be close enough to the prior set, so that $H(x_0)$ is anyway similar to $H(x_{post})$ even if the model is not linear? In addition, the authors should give a reference for Eq. 1 (e.g., Tarantola, 1987) and explictly state that because linearity is assumed it takes the formed expressed in this manuscript (while the general equation is $C_{x_{post}}^{-1} = C_{x_0}^{-1} + H(x_{post})^T C_d^{-1} H(x_{post})$).

**P7, L6:** "those observations" is at best vague and at worst confusing, since it seems to relate to "observational uncertainty" (rather than "uncertainties") but again, observational uncertainties normally also includes the model component.

**P7, L27 - P8, L8:** In this whole paragraph (and the derived results and discussion), it would be important to mention which uncertainty is dealt with (random or systematic). Since only the random error can be studied this kind of framework, the potential impact of a systematic error (a bias) should be discussed as well, or at least mentioned.

**P8, L10-11:** Any proof/reference this it is sufficient? Even if it is expert knowledge, the authors should at least state it.

**P8, L22:** "Effective constraint" rather than "constraint", might be more accurate.

**P9, L9:** Which global physiological parameters are the authors referring to? Rows 37-68 in Table A1? See earlier comment on making Table A1 clearer.

**P9, L10-17:** The values of constraints in the text do not correspond to those shown in Table A1. Please update.

**P10, L3:** Maybe add between brackets than the chlorophyll parameters are $C_{ab}$ components.

**P10, L3-4:** "During the assimilation" comes a bit abruptly. I guess the authors are talking about prospective data assimilation efforts with BETHY-SCOPE and SIF, please expand to make easier for the reader to understand.

**P10, L9:** This is a somewhat confusing formulation to say that uncertainty (and its subsequent reduction) is quantified as one standard deviation. Maybe giving this reference metric already in the methods would be helpful.

**P10, L10-15:** I suggest to have Fig. 3 (not mentioned in the text, maybe already in Sect. 2.4.?) on the color same scale as Figs. 4 and 5.

**P11, L3:** A figure showing the uncertainty reduction Could the authors briefly detail how they assessed the relative contribution of covariances to the total uncertainty in GPP? By summing the non-diagonal terms in $H_{GPP}C_xH_{GPP}^T$?

**P11, L7:** Could the authors briefly detail how they assessed the relative contribution

of covariances to the total uncertainty in GPP? By summing the non-diagonal terms in $H_{GPP}C_xH_{GPP}^T$?

**P11, L17:** As the authors state in the discussion, the fact that GPP is relatively insensitive to $C_{ab}$ derives from the lack of a mechanistic link in the model between chlorophyll content and carboxylation rate. I suggest therefore to remove the "discussive" end of this sentence here and leave for the discussion where it is explained.

**P11, L23-24:** I disagree with the last part of this sentence: it seems to me that the increase in relative uncertainty contribution of physiological processes only says that they are less constrained than other processes, therefore the stated "limitations" is just *relative* to other well-constrained parameters. Without looking at the *absolute* value of uncertainty in GPP arising from each group of parameters (from which is then calculated the relative contribution), no statement can be made about how really "limited" is the constraint of SIF in ultimately reducing the uncertainty of a given parameter to simulate GPP.

**P11, L27 to P12**: I feel that an additional figure would be needed here, to show how the constraints in GPP from given parameters groups changes across the year in Temperate and Boreal regions. It could be for example a monthly-binned boxplot, each box corresponding to the range of constraint GPP for a given group of parameters, using colors or panels to separate regions. That would help the reader to support all the description given in the main text.

**P12, L4**: "exaggerated" seems quite subjective.

**P12, L8**: The parameter $Vc_{max}$ is mentioned, then "these parameters", I guess referring to the different PFT components $Vc_{max}$? Please specify.

**P14, L10-14:** This might be suited for the discussion section.

**P15, L10:** How did the authors get this number?

**P16, L810:** I would move this sentence to the next paragraph, where diurnal dynamics are discussed.

**P15, L10:** How did the authors get this number?

**P16, L31-35:** And addtional figure showing the relative contribution of each parameters to modelled GPP uncertainty would make the results clearer. Perhaps using the same barplot setup as Fig. 1, except that y-axis would relative contribution to GPP uncertainty, and prior and posterior results could be shown using mirroring bars (2 y-axis would be needed then, one going upwards and the other downards).

**P16, L33:** "Free" sounds a bit odd here, what do the authors want to say?

**P16, L34-35:** I assumes that by "[...] only other free parameter controlling leaf area index other [...]" the authors mean that the model is highly *sensitive* to this parameter (i.e., large values in $H$), so adding to little prior *parameter knowledge* results indeed in large propagated uncertainty. The first aspect is however not quite clear from the current formulation. Since this separate consideration of *sensitivity* and *parameter knowledge* is essential when considering output uncertainty, here in the discussion I suggest detailing a bit more these aspects. Useful supporting references are, e.g., discussions in Dietze et al. (2014) and Kuppel et al. (2014).

**P17, L1-2:** This sentence ("The prevalence [...] global scale") is rather general and does not add much to the following one (which gives numbers). I suggest removing the former.

former.

**Technical comments**

**P2, L16:** Definition of NDVI and EVI acronyms, first introduced here

**P2, L23:** *has* instead of *have*.

**P2, L35:** It is not the process that provides the constraints, rather the latter being constrained!

**P6, L9:** Replaces "equation 1" by "Eq. 1". It also applies to L17, to "equation [2,3,4]" on [P6;L26], [P7;L2-L4-L14] and [P10;L8].

**P6, L10-11:** Strange formulation, I would suggest: "[. . . ] a Jacobian matrix ($H$), which is calculated around [. . . ]"

**P6, L10-11:** Strange formulation, I would suggest: "[. . . ] a Jacobian matrix ($H$), which is calculated around [. . . ]"

**P6, L26:** "p. 71" instead of "pg. 71".

**P7, L6:** "its" instead of "it's".

**P7, L10:** "described" would be more accurate than "demonstrated"

**P7, L27-29:** "As might be expected" is quite subjective. I suggest to connect the two sentences: "[. . . ] while uncertainty in forcing such as incoming radiation is not considered in the curret CCDAS setup, it is considered to be an important variable

driving SIF (Verrelst et al., 2015) and GPP (*rference needed*)."

**P9, L2:** "Table A1" instead of "Table 1".

**P10, L7:** If "as" refers only to the posterior uncertainty in GPP, it should then be replaced by "the latter being".

**P11, L12:** "stems" instead of "stem"

**P12:** "made up by" (L2) and "make up" (L8) are somewhat colloquial/vague here, it could be respctively replace by "arises from" and "contribute to".

**P14, L4:** Changing with "Second, we also increase [. . . ]" might help the reader understand you are describing the other experiment.

**P15, L7:** I suggest "[. . . ]SWRad, in both cases resulting in a relative reduction in the GPP uncertainty by about 78.6%".

**P15, L17-18:** "constraints" is repeated a lot here, I suggest: "[. . . ] ultimately yields a global annual GPP estimate within $\pm$ 2.8 PgC.yr$^{-1}$.".

**P16, L18:** "however" seems somewhat redundant.

**P16, L18:** "PSII" should be defined on L11.

**P17, L9:** "feasible with" feels odd. Maybe "acheviable using"?

**P17, L23-24:** I suggest rephrasing as follows: "This in line with Koffi et al. (2015) who found limited sensitivity of simulated SIF to $Vc_{max}$."

**P18, L7-8:** The meaning is not clear, I assumed the authors meant "While including

this forcing uncertainty increases the prior GPP uncertainty, incorporating the former within SIF uncertainty itself mitigates the downstream effect on GPP."

**P18, L16:** Maybe replace "can also be" by "will also be".

**References**

- Desroziers, G., Berre, L., Chapnik, B., & Poli, P. (2005). Diagnosis of observation, background and analysis‐error statistics in observation space. Quarterly Journal of the Royal Meteorological Society, 131(613), 3385-3396.

- Dietze, M. C., Serbin, S. P., Davidson, C., Desai, A. R., Feng, X., et al. (2014). A quantitative assessment of a terrestrial biosphere model's data needs across North American biomes. Journal of Geophysical Research: Biogeosciences, 119(3), 286-300.

- Kuppel, S., Chevallier, F., & Peylin, P. (2013). Quantifying the model structural error in carbon cycle data assimilation systems. Geoscientific Model Development, 6(1), 45-55.

- Kuppel, S., Peylin, P., Maignan, F., Chevallier, F., Kiely, et al. (2014). Model–data fusion across ecosystems: from multisite optimizations to global simulations. Geoscientific Model Development, 7(6), 2581-2597.

- MacBean, N., Peylin, P., Chevallier, F., Scholze, M., & Schürmann, G. (2016). Consistent assimilation of multiple data streams in a carbon cycle data assimilation system. Geoscientific Model Development, 9(10), 3569.

- Peylin, P., Bacour, C., MacBean, N., Leonard, S., Rayner, P., et al. (2016). A new stepwise carbon cycle data assimilation system using multiple data streams to

constrain the simulated land surface carbon cycle. Geoscientific Model Development, 9(9), 3321.

- Tarantola, A. (1987). Inverse problem theory: Methods for data fitting and model parameter estimation.

- Verrelst, J., Rivera, J. P., van der Tol, C., Magnani, F., Mohammed, G., & Moreno, J. (2015). Global sensitivity analysis of the SCOPE model: What drives simulated canopy-leaving sun-induced fluorescence?. Remote Sensing of Environment, 166, 8-21.
* * *

---

## Author Response (AR1)

**1 Author General Response**

Please find below our response to the referee comments. We appreciate the referees highly useful and constructive feedback, and have taken each comment on board to help improve this work.

Each of the referee comments are responded to by the author in blue text, including any actions taken. We also provide a general response section to address recurring comments. We would also like to add that, following on from the reviewers comments, we found one of our minor points of analysis was incorrect and has therefore been made redundant. This pertains to the analysis of scaled high-resolution observational uncertainties. This has had no effect on our conclusions.

Key changes include:

- Clarification for the calculations of the observational uncertainties.
- An additional section in the appendix showing figures as an example of how we calculated our observational uncertainties; as suggested by Referee # 1.
- Made the parameter table in the appendix much clearer and included descriptions of each one.
- Added additional figures to show the contribution of different parameter classes to uncertainty in GPP across the defined regions; as suggested by Referee # 2.
- Included an analysis of the effect of systematic errors in the observations; as suggested by Referee # 2. This includes additional figures in the appendix.
- Clarified and simplified, where possible, the explanation of aims and method of this study. Referee # 1 seemed to partly misinterpret what we were aiming to do and how we were doing it. In particular that we are not estimating parameter values, we are only assessing information content. We have therefore amended the text in the introduction and methods to make this clearer for readers.

Finally, can the editor please advise on how we put together supplementary material. We think that the Appendix sections A2 and A3 would be better suited in Supplementary material, but we are not sure how to do this. Thank you.

**1.1 Observational Uncertainty Calculations**

As pointed out by both reviewers the calculation of the observational uncertainty requires clarification. To address these recurring comments we have done the

following:

- Re-written the section in the methods on the calculation of observational uncertainties. We have gone through the calculation and justified it step by step to help readers follow what is being done and why.
- Provided a simplified equation in that section that approximates the (seemingly confusing) area-weighted uncertainty.
- Attached an additional section in supplementary material giving further details of this calculation and the exact formula. As part of this we include an example calculation for a grid cell over the Amazon with accompanying figures showing the original data and the final calculated uncertainty (for the whole 12 months).

We also clarify here. In calculating the observational uncertainties we make the assumption that the observations are independent, i.e. have uncorrelated errors. This is the same assumption made in Parazoo et al. (2013,2014).

This means, effectively, that with the aggregation of GOSAT grid cells into a larger region (i.e. the course model grid cells) there is a larger number of observations therefore the uncertainty goes down by the  $1/\sqrt{n}$  law (the same occurs when calculating the standard error). This is a well-known occurrence in dealing with satellite observations and it can be surprising to see the effect of going from single sounding precision (relatively large uncertainty) to aggregated regions (relatively low uncertainty). Another way to describe this is that if you aggregate a region you're taking many independent observations (from each sub-region) and getting out just one independent observation, so to preserve the information content of those sub-regions independent observations the uncertainty goes down; this is called the Jacobian rule of probabilities.

Characterizing correlations in errors is a known problem with satellite measurements. For SIF correlated errors may be due to, for example, error in the retrieval zero-level offset. We are currently looking into the effect of the zero-level offset and will add a additional sensitivity test in the results and discussion accounting for this. If measurements have correlated errors the information content is less than without. To be on the more conservative side we scale our uncertainties by  $\sqrt{2}$  which increases the uncertainty.

One reviewer also noted that the observational uncertainties over the tropics (and in particular the Amazon) in Figure 3 appear much smaller than expected. We recognise that this needs explaining. Amendments have been made to the methods section clarifying this, but we also clarify here. Again, the two main points above are relevant. Another element of the small uncertainty over the tropics in Figure 3 is that this is an "annual" uncertainty, so this accounts for the fact that during parts of the year the high-latitudes have no data, while the tropics almost always have data, therefore the tropics have more observations which leads to lower uncertainty.

**1.1.1 Inclusion of Structural Uncertainties**

This point relates to the calculation of the covariance matrix  $C_d$ . Formally, this is the uncertainty covariance matrix representing observational and model uncertainty. We agree that we must specify this in the methods and have thus changed it.

There are two general types of structural uncertainties.

- First, is a structural uncertainty in the model (i.e. model structural error). This may be due to incomplete process formulation in the model equations. One can address this error by looking at statistics in the model-observation mismatch following an assimilation of the data (Kuppel et al., 2013). This is therefore only feasible following an assimilation of the data to estimate posterior SIF, posterior parameters, and posterior GPP. In the present study, we are only interested in error propagation so we do not perform an assimilation of the data.
- Second, is a structural uncertainty in the observations. This may be due to certain unknown errors in space and/or time due to (for example) systematic errors in the instrument or retrieval algorithm. One example of this for SIF is an error in the zero-level offset (Frankenberg et al., 2011;2014).

We address this issue by conducting a sensitivity test. We introduce a structural uncertainty into the error propagation system to assess the effect on the calculated posterior uncertainties. We incorporate this sensitivity test into the results and discussion to approximate the effect this extra uncertainty may produce on uncertainty in GPP.

**2 Anonymous Referee # 1**

**2.1 Summary**

This paper uses satellite observations of Solar-Induced Fluorescence (SIF) in an inversion scheme (CCDAS) to reduce uncertainty in a posteriori estimates of model parameters and outputs, specifically GPP. Interestingly, no attention is given to actual parameter values or GPP estimates; the focus is entirely on how much reduction in uncertainty can be expected due to the inclusion of SIF.

The paper is reasonably well written, and uses a novel approach to attempt to reduce uncertainty in a posteriori estimates of model parameters and output. However, I feel that the paper needs clarification and perhaps some reorganization to help readers to follow the story. Furthermore, I believe that the critical issue of observational uncertainty is given too little attention and must be clarified.

The authors provide reasonably comprehensive citations for CCDAS, but the paper reads is if it were written (as it probably was) by someone who is a Data Assimilation (DA) expert. To this reviewer it seems that some details are either implied or 'skipped over'. It is likely that many readers will be DA experts themselves, but the inclusion of SIF will probably draw in readership that may not possess the DA expertise to easily understand what is going on. I may be a member of that part of the audience, so some clarification is warranted. Specifically, the relationship between covariance matrices (Cx, Cd) and standard deviation () is not entirely clear. Good point. We want readers from different audiences to be able to follow what was done easily. We have added and clarified text in the methods section to help non-DA readers relate covariance matrices to standard deviation uncertainty simplified other points where possible. We have also modified the last paragraph of the introduction to make it clearer what the specific aims are.

The description of grids used and observation area ("GOSAT grid cell"; section 2.4) needs clarification. Two grid sizes are mentioned in Section 2.4, but we don't learn much more about them until Section 2.5. Good point. We have amended this as suggested by shifting the grid resolution information to the beginning of section 2. I would like to see a more deliberate explanation of "here is what we are going to do, and here is how we are going to do it". That might fit better in Section 2.1. Some specific Issues:

• Figure 3, showing observational uncertainty, is not referred to in the section describing observational uncertainty. It needs to be. Amended.

**2.2 Observational Uncertainty**

Eqn 4: I see two ways that this value can be small: 1) there are many observations, and  $\sigma^2$  is small. 2) There are very few observations, and Area is small. Parazoo et al. (2013) estimated uncertainty as the standard error. This has the effect of allowing a large error in regions with very few observations, like the tropics. Figure 3 in the manuscript under review shows some of the smallest observational uncertainty in the tropics, and that makes absolutely no sense to me. I've worked with the GOSAT data, and over the deepest tropics there are very few observations, which makes me suspicious that your uncertainty is small because of reason 2). Parazoo et al. did not extend their analysis to the wetter parts of Amazonia because they just didn't have enough data to justify it. Now the authors claim that this region has some of the smallest observational uncertainty on the globe! A detailed justification of how uncertainty can be very small over a region with few or no datapoints is an absolute necessity. Please refer to general response section above.

I do not think multiplying by square-root-2 is sufficient to remedy what might be unrealistically low uncertainty values. Please refer to general response section above.

When GOSAT 2010 data is aggregated onto the 1.25x1.0 degree MERRA grid, I see that the maximum number of retrievals for a given month, anywhere on the globe, is between 30-35 or so. Looking at South America, I see that very few MERRA gridcells have more than 10 retrievals in a given month during 2010, and many gridcells have 5 or fewer. Aggregating up to 7x10 (or 2x2) you are not going to get very much increase in sample size. Id like to see the authors address the sparseness of the GOSAT data and explain how this will or will not effect their method. In the amended manuscript we show an example calculation over a 7.5x10 degree grid cell including the GOSAT sub-grid cells to show how this scales across 2010. We see that aggregating from 3x3 to 7.5x10 you get actually see a big increase in sample size. For example in Jan 2010 any GOSAT sub-grid cell may have between 0-20 soundings, but aggregating to the 7.5x10 there is almost 80.

The number of GOSAT observations is invariant and does not change with grid size. The aggregation of GOSAT observations changes with grid size (Section 2.4). This should be clarified. In fact, the number of GOSAT observations does vary with grid size. With a larger grid size you capture more GOSAT soundings. You may refer to the general response section for further details. We have clarified this in the methods section.

An individual GOSAT retrieval has pixel size of around 10 km2, I believe. OCO-2 will have a pixel size of  $\sim 5 \text{ km}^2$ , and GOME-2 is a 40-80 pixel, or 3200 km2. This will have a large impact on your inversion scheme and the calculation of observational uncertainty. Since this paper only uses GOSAT, the other products probably dont need too much (or any?) explanation, but I do have questions about GOSAT and the grids used:

- 1. There is the possibility for (possibly) many 10km2 GOSAT retrievals to be included in a 7.5x10 degree gridcell. For that matter there can be many of them in a 2x2 gridcell too. BETHY-SCOPE tiles in 3 PFTs; how are GOSAT retrievals registered to these PFTs? This is a good point. The observations are not separated per PFT, doing so would effectively triple the information content as there would be three times more observations, which would in fact improve the results. We compare observations at the grid cell scale. Thus information is transferred/split to PFTs through the Jacobian sensitivities, which account for PFT fractions. E.g. if a grid cell is 90% C3Gr, then the SIF sensitivity over that grid cell will be dominated by parameters relating to C3Gr, with smaller contributions from the PFTs that make up the remaining 10%. Thus, the information content of the observations is split accordingly. Are GOSAT retrievals marked with a specific land cover type, and accumulated on a per-PFT basis? What about GOSAT retrievals that are not associated with one of the 3 PFTs tiled into the BETHY-SCOPE gridcell? Are they discarded? Why or why not? We do not attempt to disaggregate observations in this way. We assume there is roughly even coverage across the PFTs, even though the absolute footprint of a GOSAT sounding is about  $10 \text{km}^2$ , it has a wide swath of around  $750 \text{ km}^2$  with 5 footprints. Thus we assume decent coverage. This will be more important to consider in a full assimilation of the data i.e. for estimating parameter values and fluxes.
- 2. If all GOSAT retrievals within a gridcell are utilized, is the mean taken and used for DA with all 3 PFTs? In this case arent you 'smearing out' the information that SIF provides? Guanter et al. (2012) demonstrate that the linear relationship between SIF and individual PFTs is heterogeneous. Do you take this into account? If so, how? If not, why not? This is true for a full assimilation and parameter estimation but in this study, we do not consider the mean values of the observations, only their uncertainties as we're only interested in information content. Thus these issues are not present.
- 3. In August 2010 the GOSAT scan strategy was changed; the area observed was decreased, but the number of retrievals over a given region was increased. How does this effect the two questions above? Yes, good point. The observational uncertainties used in section 2.4 are standard errors (although slightly adjusted to increase the uncertainty as described in section 2.4), thus they account for the number of observations per grid cell.

The reduction in uncertainty for global GPP is dramatic (79%). However, this reduction is critically dependent upon Cd (observation uncertainty) according to equation 1. Therefore, I think it is absolutely essential that the questions surrounding the determination of this observation uncertainty are answered in a clear and categorical manner. Agreed. We have clarified our calculation of the observational uncertainties in the manuscript. Please refer to general response

**above.**

Im not a DA expert, but I do collaborate with quite a few people who are, and I think I understand the basics. The covariance matrices are absolutely fundamental to the outcomes of a DA experiment: If the observational uncertainty is small and the model uncertainty large, the a posteriori outcome can be pulled strongly towards the observations. If the opposite is true, then it will be hard to budge the inversion away from the model prior. Is this correct? Essentially, yes this is true. However, we note that this question primarily applies to a an assimilation with real data. In this paper we assess the information content of SIF observations, i.e. only uncertainties of model parameters and GPP, not their values. We can do this because this is a linear problem, whereas the full assimilation is a non-linear problem and the subject of subsequent study. The point regarding observational uncertainty vs model uncertainty is pertinent however, and we address this in the general response section.

In this paper the first case is presented: the observational uncertainty is, to my eye extremely small and therefore results in an amazing reduction in uncertainty in the a posteriori result.

The absence of evaluation of actual posterior values of either parameter or flux values may actually hinder the analysis. If the result of the study is an outlandish value for global GPP, then that might indicate a problem. Of course, estimates of global GPP vary by about a factor of two (Huntzinger et al., 2012), so maybe this wouldnt help as much as one might hope. However, posterior parameter and flux values might offer insight, and a comprehensive evaluation of method and results (values of parameters and flux) could provide more support for the authors' conclusions. Was this considered? Why or why not? Im suspicious that posterior flux and parameter values were outlandish, and a choice was made to focus on method even though results may be untrustworthy. I suspect many readers will have this suspicion too. Assessing information content of the observations is a linear problem which can be performed independently of comparing actual values of model and observed data. This is convenient as an assessment of the information content tells us whether SIF is going to be a useful constraint on GPP before we have to go through the challenging process of fully assimilating the data. We also believe that the information content study here is substantial enough. Adding in a full assimilation to estimate parameter and GPP values is a complicated non-linear problem and adding this into the current manuscript would make for too large a study. An assimilation of the data where one actually estimates global GPP is the subject of subsequent study.

A detailed description of the construction of the observation uncertainty may detract from the papers readability, but including it in an appendix would be appropriate. Additionally, I would like to see, perhaps in supplemental material, a step-by-step description of the calculation of the observation uncertainty, perhaps in the 7x10 gridcell that contains Manaus, Brazil. Agreed. Refer to general response section. To see such a large reduction in error sent warning bells ringing with me; I dont think it is an overstatement to say that the entire paper depends on the observation uncertainty. If the authors can demonstrate that the values shown in Figure 3 are justifiable, then the paper has merit. If not, I think the whole endeavor falls apart, as the structural underpinning would have disintegrated. In that case the paper is not worthy of publication.

**2.3 Specific Comments**

- Figure 2: The information here is too dense (small labels, tiny resolution on the plot) to follow. If the only pertinent information is in the lowerright- hand of the plot, why not omit the rest and enlarge this sector of the graph? Good point. We have edited this figure to make it clearer. We have removed any rows/columns that have no correlations and increased the font size.
- Figure 2: There is very little description of the graph and what it means. Again, this may be another case where the authors are assuming that their readers look at graphs like this every day and know what it is showing. Yep fair enough. We've provided a better description in the text and caption.
- Figure 3: What are the units? Amended.
- Figure 4: Absolute uncertainty annual GPP will of course correlate directly with productivity. If you standardize the time series and look at relative uncertainty I imagine that map will look very different. Have you done this? If you have, do Figures 4-6 look similar or different? We need some clarification here from the reviewer. We can do the following: prior uncertainty divided by prior GPP and posterior uncertainty divided by prior GPP. But we cannot do the following: posterior uncertainty divided vided posterior GPP. As this is an error propagation study we have not estimated posterior GPP.
- Table 1A: There is no description of what these parameters are and what they do. There are sporadic mentions in the text, but for the most part the reader is left to ones self to figure out what these parameters are for. I would like to see a column added (there appears to be room, as the uncertainty reduction columns could be re-formatted) with a couple of words or a phrase describing each variable. Section 3.2: line 14 on page 11 mentions that  $\tau_W$  makes up 82% of the global annual uncertainty in posterior global GPP. The reader does not know what W is. At the end of Section 3.1 there are several other parameters listed, and again the reader is not told what they are. It might be helpful to have a short description in parentheses following the listing of each parameter, but I would prefer to see that information in table 1A. Good point. We will amend Table A1 and make it clear what the parameters mean if referring to them in text.

- Boilley and Wald (2015) discuss a high bias in the radiation from reanalyses. Im not sure this is the same as the uncertainty mentioned in sections 2.4 and 3.4. Can you elaborate? We were not aware of the Biolley and Wald (2015) study, so we thank the reviewer for the citation and we have included it in the manuscript. A known bias in the radiation such as this should be removed from the reanalyses data before it is distributed. We consider an uncertainty of unknown sign, as shown in Kato et al. (2012), which can be accounted for in the prior uncertainty and constrained through the error propagation system as we demonstrated. We clarify this in the manuscript.
- Page 17, lines 7-8: "...we can predict and quantify how SIF wil constrain the uncertainty of process parameters and GPP, but we cannot predict how their values will change". Why not? Can't you back the posterior values out of the a posteriori covariance matrices and the Jacobian? Isn't the whole point of DA to obtain these posterior values? The process of getting posterior parameter values and obtaining posterior fluxes is a nonlinear problem that is therefore arduous and challenging. So, before one goes down this path they can actually assess whether it is worthwhile doing by first assessing the information content, this is linear problem and therefore simpler. However, as SIF has not been used in a full DA system with a process-based model like this before it is valuable to show, in detail, what SIF may constrain, how it does it, and any caveats to this. It seems we have not made it clear enough exactly what this study is and exactly why we are doing it. We have therefore added in some extra points to the introduction and methods section to clarify this.

**3 Anonymous Referee # 2**

**3.1 Summary**

This study evaluates the benefit of assimilating satellite-retrieved chlorophyll fluorescence into a mechanistic land surface model, to reduce the uncertainty in model parameters and simulated gross primary production (GPP). This study indeed tackles a critical issue in the current efforts towards making the most of diverse data infromation content when building efficient carbon cycle data assimilation systems.

There are, however, a few important issues in this manuscript, some of them critical. They are listed in the general comments below, followed by specific remarks/corrections.

**3.2** General Comments**

First, while the manuscript is often fairly written, on numerous occasions sentences are redundant, strangely formulated, thus logical progression of arguments is hard to follow. Frankly, it sometimes feels as if the authors did not read themselves again before submitting the manuscript. It could be just be a matter of style, but in some occasions it simply results in a lack of clarity. While I tried to list specific parts in the Specific comments and Technical comments section, I suggest a strong effort of rewriting in general. That will also make the manuscript much more accessible to modellers/data experts outside the field of CCDAS or even data assimilation at large.

Second, and perhaps more importantly, the way the observation uncertainty used in Eq. 1 is defined is quite vague. Judging from the elements presented in Sect. 2.4, it seems that only the measurements uncertainty of GOSAT retrievals of SIF is accounted for in CD, neglecting the structural uncertainty (CT, using the notation of Tarantola (1987)) of the BETHY-SCOPE model. If structural uncertainty is considered, that should be detailed in Sect. 2.4. If CT is not taken into account, this would bear important consequences. While CT is hard to estimate explicitly (although some diagnostic methods exist, e.g. see Desroziers et al. (2005), applied to land surface models by Kuppel et al. (2013)), its magnitude and structure might be comensurate or even dominant over measurement uncertainties when building CD. Not including it in Eq. 1 would then largely underestimate the posterior uncertainty of parameters and, by propagation that of modelled GPP. As noted for another reviewer, this would constitutes a serious theoretical flaw in the scope of this study and make it unsuitable for publication. Please refer to general response section.

**3.3** Specific Comments**

P2, L11-12: This sentence is rather vague, can the authors be more precise and add references to support this assertion? We have re-worded this and provided references.

P2, L27-28: Data assimilation is not only used with mechanistic models nor for ter- restrial carbon cycle modeling. I suggest to reformulated, for example: In the case of mechanistic models, this is done by constraining the simulated underlying processes. Good point, we have amended this as suggested.

P2, L28-32: In this review of the state of the art, efforts from other groups to build mechanistic CCDAS might deserve to be cited as well, e.g. (Peylin et al., 2016) and the discussion/review by MacBean et al. (2016). Absolutely. We have amended this in the manuscript.

P3, L4-6: Some references would be necessary to back these assertions. We have added references to these points.

P5, L8: The last sentence of this paragraph feels rather clumsy, it should reformulated. We have re-written this last sentence.

P5, L9: Table A1 is rather long and that is fair game given the number of parameters, yet to make it more reader-friendly I would suggest to:

- include a description column for each type of parameter,
- add the corresponding PFT between brackets for all PFT-dependent parameters, as is done for Vcmax,
- add subsection rows with parameter categories (leaf growth, ecophysiology etc.).

Good point. We have updated the table as suggested.

P6, L2-3: It is because the PDFs of parameters and observations is treated as Gaussian that it can be described by their first two moments, mean and standard deviation (taken here as the metric of uncertainty, that might need to be specified here already well), not the other way around. Yes that is correct, we have amended this.

P6, L1-4: The definition of observations here should be more precise; the reader (especially if not familiar with the data assimilation vocabulary) would assume it relates to measured observations (as the previous paragraph uses SIF observations to designate measurements), while in a rigorous probabilistic framework it should refer to quantities in the observation space (including measurements and model outputs, see General comments). We thank the reviewer for the clarification. This section has been more explicit here to make it clear what observational information is, in particular reference to  $C_d$ .

P6, L12-13: I guess that the authors meant with this sentence that a) in a linear

world H is independent from x, but b) this is an oversimplication, therefore c) bringing limitation in accuracy to a method relying on H(x0) to approximate H(xpost). It is not clear at all from the current formulation, which even almost suggest that because of linearity the choice of x0 can influence the results (through a changing H)... We thank the reviewer for pointing out this possible misinterpretation. This has been re-formulated to ensure it is clear.

P6, L13-21: I am not sure how the use of prior knowledge limits the effect of this problem: is it because we assume that the posterior parameters values will be close enough to the prior set, so that H(x0) is anyway similar to H(xpost) even if the model is not linear? In addition, the authors should give a reference for Eq. 1 (e.g., Tarantola, 1987) and explicitly state that because linearity is assumed it takes the formed expressed in this manuscript (while the general equation is  $C_{x_{post}}^1 = C_{x_0}^1 + H(x_{post})^T C_d^1 H(x_{post}))$ . Yes, in part it is because we assume  $x_0$  is close the the global optimum that would be obtained in a full assimilation i.e.  $x_{posterior}$ . Considering the parameter space is vary large, the use of prior knowledge places the parameters into a reasonable physical range. Subsequently the sensitivities calculated in H are more reasonable. We also assume that these functions are smooth. The BETHY-SCOPE model also has no step functions, (which would cause large differences in H even for a slightly different  $x_0$ ). In fact, even if there were step functions, Knorr et al. (2010) points out that a population of plants that, individually, have step functions, average up to a smooth function across a grid cell. We thank the reviewer for the clarification, we have amended the methods section to reflect this point.

P7, L6: those observations is at best vague and at worst confusing, since it seems to relate to observational uncertainty (rather than uncertainties) but again, observational uncertainties normally also includes the model component. We have re-written this sentence to be more precise.

P7, L27 - P8, L8: In this whole paragraph (and the derived results and discussion), it would be important to mention which uncertainty is dealt with (random or systematic). Since only the random error can be studied this kind of framework, the potential impact of a systematic error (a bias) should be discussed as well, or at least mentioned. Agreed. The error we can consider is a random error of unknown sign, which would still in fact be systematic as we apply a scaling factor to all of the forcing data. As another reviewer pointed out, we do not consider a known bias (i.e. systematic error of known sign) as this should be corrected for in the data already. We have now clarified this section.

P8, L10-11: Any proof/reference this it is sufficient? Even if it is expert knowledge, the authors should at least state it. Perhaps sufficient is the wrong word here. In using a low-resolution grid, this model equations are the same as a high resolution grid such that H relates SIF and GPP to parameters in effectively the same way. And considering Gaussian uncertainties propagate linearly in this study (i.e. with associated assumptions), the model grid resolution does not matter so much. We have re-worded this. P8, L22: Effective constraint rather than constraint, might be more ac- curate. Yes, this might be more accurate. We have changed constraint to effective constraint where ever necessary.

P9, L9: Which global physiological parameters are the authors referring to? Rows 37-68 in Table A1? See earlier comment on making Table A1 clearer. We have amended table A1 as suggested so this should be more clear now.

P9, L10-17: The values of constraints in the text do not correspond to those shown in Table A1. Please update. Good catch! Thanks. We have updated these values.

P10, L3: Maybe add between brackets than the chlorophyll parameters are  $C_{ab}$  components. Yep, thanks. This should be shown as Cab rather than worded chlorophyll, so we have amended this.

P10, L3-4: During the assimilation comes a bit abruptly. I guess the authors are talking about prospective data assimilation efforts with BETHY-SCOPE and SIF, please expand to make easier for the reader to understand. We have amended this to say Thus, during a full assimilation with the SIF data only the sum... as is done in other parts of the paper.

P10, L9: This is a somewhat confusing formulation to say that uncertainty (and its subsequent reduction) is quantified as one standard deviation. Maybe giving this reference metric already in the methods would be helpful. Okay. We have added in this reference metric to the methods section under Uncertainty Calculations.

P10, L10-15: I suggest to have Fig. 3 (not mentioned in the text, maybe already in Sect. 2.4.?) on the color same scale as Figs. 4 and 5. We have now referred to it in the text. Figures 3 and Figures 4/5 are different quantities (SIF and GPP, respectively) so we dont think they need to be on the same scale. However, we note that better labeling is required for these figures to make it clear theyre different quantities, so we have done this.

P11, L3: A figure showing the uncertainty reduction Could the authors briefly detail how they assessed the relative contribution of covariances to the total uncertainty in GPP? By summing the non-diagonal terms in H GPP  $C_x$   $H^T$ ? GPP Using equation 3 we assess the constraint with the full covariance matrix  $C_x$  (i.e. including off-diagonal terms). Then we assess the constraint with off-diagonal terms set to zero in  $C_x$ . The difference between these two cases is the contribution of correlations. We have now outlined this in the manuscript.

P11, L7: Could the authors briefly detail how they assessed the relative contribution of covariances to the total uncertainty in GPP? By summing the nondiagonal terms in  $H \ GPP \ C_x \ H^T$ ? Yes, we have added in an extra sentence GPP explaining what we did as follows we can assess the contribution of these correlations to the constraint of GPP by setting all off-diagonal elements in Cxpost to zero in Equation ??, the difference between this and the standard case that uses the full  $C_{x_{nost}}$  equates to the contribution of correlations.

P11, L17: As the authors state in the discussion, the fact that GPP is relatively insensitive to Cab derives from the lack of a mechanistic link in the model between chlorophyll content and carboxylation rate. I suggest therefore to remove the discussive end of this sentence here and leave for the discussion where it is explained. Okay, good point.

P11, L23-24: I disagree with the last part of this sentence: it seems to me that the increase in relative uncertainty contribution of physiological processes only says that they are less constrained than other processes, therefore the stated limitations is just relative to other well-constrained parameters. Without looking at the absolute value of uncertainty in GPP arising from each group of parameters (from which is then calculated the relative contribution), no statement can be made about how really limited is the constraint of SIF in ultimately reducing the uncertainty of a given parameter to simulate GPP. That is correct and a good point to make. We have amended this statement to say Uncertainties in physiological parameters are constrained less than the leaf growth parameters which results in them contributing more in relative terms to the posterior uncertainty of GPP.

P11, L27 to P12: I feel that an additional figure would be needed here, to show how the constraints in GPP from given parameters groups changes across the year in Temperate and Boreal regions. It could be for example a monthlybinned boxplot, each box corresponding to the range of constraint GPP for a given group of parameters, using colors or panels to separate regions. That would help the reader to support all the description given in the main text. Okay, good suggestion. We have added another figure here to show the contributions of parameter groups to uncertainty across the year for each region.

P12, L4: exaggerated seems quite subjective. Okay, we have re-phrased this to say Similar differences between seasonal constraint is seen for the Temperate North, although with a smaller seasonal variation in SIF constraint that ranges between 74% and 87% across the year.

P12, L8: The parameter Vcmax is mentioned, then these parameters, I guess referring to the different PFT components Vcmax? Please specify. In fact were referring to the  $\tau_W$  parameters, so we have now specified in the manuscript.

P14, L10-14: This might be suited for the discussion section. Agreed. Amended.

P15, L10: How did the authors get this number? As its effectively treated as a parameter, we can assess the relative uncertainty reduction by the same equation used for parameters (i.e.  $1-\sigma_{post}/\sigma_{prior}$ ). We have specified how this is done in the manuscript now.

P16, L810: I would move this sentence to the next paragraph, where diurnal dynamics are discussed. Amended.

P16, L31-35: And additional figure showing the relative contribution of each parameters to modelled GPP uncertainty would make the results clearer. Perhaps using the same barplot setup as Fig. 1, except that y-axis would relative contribution to GPP uncertainty, and prior and posterior results could be shown using mirroring bars (2 y-axis would be needed then, one going upwards and the other downards). This is a good point, an additional figure will help readers follow results+discussion. We have created a figure similar to described: with classes of PFTs and their contribution to uncertainty in GPP (in Pg C yr-1) across the year. We thank the reviewer for this suggestion.

P16, L33: Free sounds a bit odd here, what do the authors want to say? Just that it is the only process parameter that is optimizable (i.e. not a fixed parameter). We have removed free and changed this to be process parameter.

P16, L34-35: I assumes that by [. . . ] only other free parameter controlling leaf area index other [. . . ] the authors mean that the model is highly sensitive to this parameter (i.e., large values in H), so adding to little prior parameter knowledge results indeed in large propagated uncertainty. The first aspect is however not quite clear from the current formulation. Since this separate consideration of sensitivity and parameter knowledge is essential when considering output uncertainty, here in the discussion I suggest detailing a bit more these aspects. Useful supporting references are, e.g., discussions in Dietze et al. (2014) and Kuppel et al. (2014). This is helpful. We have clarified this in the manuscript.

P17, L1-2: This sentence (The prevalence [. . . ] global scale) is rather general and does not add much to the following one (which gives numbers). I suggest removing the former. Agreed. Amended.

**3.4 Technical Comments**

P2, L16: Definition of NDVI and EVI acronyms, first introduced here. Amended.

P2, L23: has instead of have. has doesn't sound right to me.

P2, L35: It is not the process that provides the constraints, rather the latter being constrained! Amended.

P6, L9: Replaces equation 1 by Eq. 1. It also applies to L17, to equation [2,3,4] on [P6;L26], [P7;L2-L4-L14] and [P10;L8]. Amended.

P6, L10-11: Strange formulation, I would suggest: [. . . ] a Jacobian matrix (H ), which is calculated around [. . . ] Amended.

P6, L26: p. 71 instead of pg. 71. Amended

P7, L6: its instead of its. Amended.

P7, L10: described would be more accurate than demonstrated. This section has been re-written, demonstrated is no longer present.

P7, L27-29: As might be expected is quite subjective. I suggest to connect the two sentences: [. . . ] while uncertainty in forcing such as incoming radiation is not considered in the curret CCDAS setup, it is considered to be an important variable driving SIF (Verrelst et al., 2015) and GPP (rference needed). Amended.

P9, L2: Table A1 instead of Table 1. Amended.

P10, L7: If as refers only to the posterior uncertainty in GPP, it should then be replaced by the latter being. It refers to both prior and posterior, so we have left this as it is.

P11, L12: stems instead of stem. Amended.

P12: made up by (L2) and make up (L8) are somewhat colloquial/vague here, it could be respectively replace by arises from and contribute to. Amended.

P14, L4: Changing with Second, we also increase [. . . ] might help the reader understand you are describing the other experiment. Good point, amended.

P15, L7: I suggest [. . . ]SWRad, in both cases resulting in a relative reduction in the GPP uncertainty by about 78.6%. Amended as suggested.

P15, L17-18: constraints is repeated a lot here, I suggest: [. . . ] ultimately yields a global annual GPP estimate within 2.8 PgC.yr1.. We have altered this sentence already to specify that it is parametric uncertainty that is reported. It reads: ...and ultimately yields a parametric uncertainty in global annual GPP of 2.8 PgCyr1

P16, L18: however seems somewhat redundant. Agreed. Amended.

P16, L18: PSII should be defined on L11. Amended.

P17, L9: feasible with feels odd. Maybe acheviable using? Amended.

P17, L23-24: I suggest rephrasing as follows: This in line with Koffi et al. (2015) who found limited sensitivity of simulated SIF to Vcmax. Amended.

P18, L7-8: The meaning is not clear, I assumed the authors meant While including this forcing uncertainty increases the prior GPP uncertainty, incorporating the former within SIF uncertainty itself mitigates the downstream effect on GPP. Yes, this sentence is a little confusing. We have re-worded in the manuscript similarly to suggested.

P18, L16: Maybe replace can also be by will also be. Amended.

**Assimilating solar-induced chlorophyll fluorescence into the terrestrial biosphere model BETHY-SCOPE v1.0: Model description and information content**

Norton Alexander J.1, Rayner Peter J.1, Koffi Ernest N.2, and Scholze Marko3

1School of Earth Sciences, University of Melbourne, Australia

2European Commission Joint Research Centre, Ispra, Italy

[revised manuscript text omitted]

Where feasible, systematic uncertainties in the SIF observations should also be considered in error propagation analyses. While systematic errors in the model cannot be assessed prior to a full assimilation of the data (Kuppel et al., 2013), systematic errors in the observations can be. To incorporate this into our analysis, we investigate one source of structural uncertainty

- 5 due to potential errors in the zero-level offset. The zero-level offset correction is done to prevent biases in the SIF retrieval (Frankenberg et al., 2011a). Based on previous analyses, systematic uncertainties in the SIF retrieval may be considered small (Frankenberg et al., 2011a, 2014). Here, we provide a more detailed assessment and characterization of the in-orbit systematic uncertainties. This is performed by assessing zero-level offset corrected GOSAT SIF soundings over the non-fluorescent regions of Antarctica and central Greenland during January and July, respectively. Systematic errors appear quite small (±
- 10  $0.06 W m^{-2} \mu m^{-1} sr^{-1}$  (see Appendix Figure 10) and may vary seasonally. We therefore assess the effect of a conservative systematic random error of size  $\pm 0.1 W m^{-2} \mu m^{-1} sr^{-1}$  in the zero-level offset seasonally. This provides a sensitivity test of incorporating this systematic uncertainty into the error propagation system.

An additional source of uncertainty in model estimates of GPP is climate forcing. As mentioned by Koffi et al. (2015), while uncertainty in forcing such as incoming radiation is not considered in the current CCDAS setup. As might be expected

- 15 however, it is considered to be an important variable in driving SIF (Verrelst et al., 2015) and GPP (Farguhar et al., 1980). Without consideration of uncertainties in forcing variables the uncertainty in GPP may be underestimated. Studies that use process-based models or empirically-derived relationships do not explicitly consider such uncertainties (e.g. Beer et al., 2010). One such forcing variable is downward shortwave radiation (SWRad). Monthly means of SWRad are suggested to have an uncertainty a random error of 12  $Wm^{-2}$  due mostly to uncertainty in clouds and aerosols (Kato et al., 2012). We therefore
- 20 investigate SWRad uncertainty may be considered in GPP estimates. Furthermore, as SIF responds strongly to SWRad, there is the potential to utilize SIF observations as a constraint on the uncertainty of the forcing. We therefore conduct an additional experiment that incorporates the uncertainty in SWRad in the error propagation system. For this experiment an additional parameter representing SWRad is added to the inversion, which acts as a scaling factor for SWRad globally. We investigate the level of constraint SIF provides on this scaling factor, and the subsequent effects of incorporating uncertainty in SWRad in
- 25 this inversion on uncertainty in GPP.

**2.5 Model and Data Setup**

In this study BETHY-SCOPE is run for the year 2010 on a computationally fast the computationally efficient, low-resolution grid scale spatial grid ( $7.5^{\circ} \times 10^{\circ}$ ), sufficient for investigating error propagation. 
[revised manuscript text omitted]

---

## Author Response (AR2)

**1 Author General Response**

We thank the reviewers once again for their valuable feedback and contribution to this study. Our specific responses to reviewer comments are shown further below in blue.

Through this review process we have noticed some additional points we consider worth correcting.

First, is that calculation of the observational uncertainties for the high-resolution case (2 x 2 degrees) are not valid. This is because we effectively preserve the information content in the GOSAT observations (3 x 3 degrees) when calculating the error reduction in the main experiment. Therefore, the additional test case experiment where we scaled the observational uncertainties by $1/\sqrt{19}$ has been omitted.

Second, is a follow up on the issue we already flagged with the editor and editorial support. As flagged, we found some inconsistencies in the model that required some additional investigation. These were found to relate to the biochemistry of the model code. The inconsistencies have now been reconciled, the analysis has been run again, and the manuscript has been updated with the new calculated values. Overall, the main conclusions of the paper have not changed and the effective constraint from SIF on global GPP has changed little (from 79% to 73%). The predicted global GPP, however, increased following this correction and therefore the reported prior and posterior uncertainties (in PgC yr-1) increased. The corrections made meant that some biochemical parameters were also omitted from the analysis. The total number of parameters has gone from 72 to 53. This correction is more consistent with the theory underlying the Collatz biochemical models of C3 and C4 photosynthesis and specific for the SCOPE calculation of GPP.

**2 Anonymous Referee # 1**

Note: Author comments are shown in blue.

First of all, I thank the authors for their effort in addressing the referee's comments, and indeed the manuscript is now much clearer. However, I am still not convinced by the way the authors circumvented the issue of estimating the total observational error. This is why I suggest to revise it before being considered for publication, and below is a more detailed account.

The authors claim in section 1.1.1 of their response, and as a reason for leaving aside the structural/model uncertainty in posterior error propagation, that this uncertainty cannot be estimated prior to performing a full assimilation. This is simply not true. One of the diagnostics provided by Desroziers et al. (2005) calculates the observational uncertainty (model + measurements) as the difference between prior model-data mismatch and propagated prior parameters error (Eq. 1 therein), which is one of the method put into application by Kuppel et al. (2013) (see Eq. 1 therein). Therefore, nothing prevents the authors from using this method. All the more because neglecting the model error in the covariance matrix $C_D$ could significant impact the large uncertainty reduction estimates currently presented in the manuscript. All the more that this study presents a novel couplings of model components where the structural uncertainty is essentially unknown. Ignoring this central aspect seriously undermines the impact of this paper.

Thank you for your feedback and indeed your are correct. We have performed this analysis as described below and included it in the manuscript. We have added the equation given below in the methods section of the manuscript, and added this analysis into the results and discussion.

We calculated the so-called Bayes' factor, or a reduced $\chi^2$ metric ($\chi_r^2$). We do not believe it is appropriate to perform this using the low-resolution version of the model as is done throughout the rest of the manuscript considering the large contribution of representation errors. Therefore, we conducted a forward run at a higher resolution (2x2 degrees), more credible for this purpose. This simulation has been brought forward from some analyses from unpublished work at this higher resolution using SIF from the OCO-2 satellite for the year 2015. While there is a slightly different discretization of the parameters in this version, it provides an insight into whether our uncertainties are consistent with the model-data mismatch. The formulation is follows:

$$\chi_r^2 \;=\; \frac{1}{N}\,(M_{x_o} - d)(H C_{x_o} H^T + C_d)^{-1}(M_{x_0} - y) \tag{1}$$

where N is the number of observations, $M_{x_o}$ is the model simulated SIF using prior parameter set, and $d$ is the SIF observations. From this we get a $\chi_r^2$ equal to 0.97, which gives evidence that our assumptions around uncertainties fit with the model-data mismatch.

On page 8, lines 21-24, it seems to me that there is a confusion between structural and systematic errors; a probabilistic framework (such as that used by Kuppel et al. (2013)) can only characterize random errors (as opposed to systematic ones). Thus nothing really can be stated about systematic errors in the model, and it cannot be opposed to the systematic errors of measurements. Note that, as stated in the original report, using 'observations' on line 23 is rather confusing, data/measurements would be more appropriate. In addition, I don't understand what kind of 'structural uncertainty' the authors are talking about on line 24, since just before they talk about systematic error in observations; did you mean 'systematic error'? As stated above, I think this section requires further analysis.

Thank you for the comment. First, our terminology is perhaps not clear enough here, thank you for bringing that to our attention. We have modified the manuscript such that there is clear distinction between "measurement/data error" and "observation error". Additionally we provide a clear description of what we mean by a "systematic error". Second, as you point out, a probabilistic framework can only characterize random errors. A distinction can be made between a systematic error and a bias. We consider a systematic error to be a randomly distributed error of unknown sign that applies to a subset of the measurements. The zero-level offset is this type of error. A bias is a systematic error with a known sign hence it cannot be included in this framework. This can be denoted as follows, where the true value $(y_i^t)$ of a quantity (e.g. SIF) can be given by

$$y_i^t = y_i + \varepsilon_i + z \tag{2}$$

where $y_i$ is the measured value at index point $i$, $\varepsilon_i$ is a random variable of which the variance is equal to $\sigma^2$, and $z$ is a random variable that has some variance and is constant for a subset of the measurements (e.g. of a particular region or time). In this sense $z$ is a systematic error. In the case of the zero-level offset in the GOSAT SIF measurements, $z$ is a systematic error of unknown sign and applies to the whole globe each month. We can approximate the magnitude of this error, but in reality we do not know it's sign. This means we can assess the effect of $z$ on the error propagation analysis from SIF to GPP. We clarify this in the methods section of the manuscript.

[revised manuscript text omitted]

---

## Author Response (AR3)

**1 Author General Response**

We thank the reviewers once again for their valuable feedback and contribution to this study. Our specific responses to reviewer comments are shown further below in blue.

Through this review process we have noticed some additional points we consider worth correcting.

- We have added a small clarification that we saw was necessary in the methods section "Uncertainty in Observations and Model Forcing Variables".
- We have made some minor grammatical changes.
- We have made some additional changes (not mentioned further below) in the discussion section to improve readability.

**2 Anonymous Referee # 1**

**Note: Author comments are shown in blue.**

First, regarding the observation error, which includes data, model (and representation) errors. I agree the  $\chi^2_r$  test is a good way to assess the observational error is well defined. However, for the optimisation problem at hand, I have three comments:

1. this test is chi-squared per degrees of freedom:  $\chi_r^2 = \chi^2/N$ , where N is the number of degrees of freedom, i.e., number of observations minus number of parameters (e.g., Taylor, 1997). By including only the formed, the authors might thus underestimated the value of  $\chi_r^2$ . Thank you for the comment. The comment is accurate except that the prior parameters are also independent pieces of information (i.e. degrees of freedom) (see introduction section of Michalak et al., 2005, doi: 10.1029/2005JD005970, and; see Tarantola and Vallette, 1982). Therefore we must add the number of observations with the number of prior parameters, then minus the number of estimated variables as you point out (i.e. posterior parameters). The result is that the prior and posterior parameters cancel and the number of degrees of freedom is equal to the number of observations. In any case, the number of observations at a high-resolution (in our case 2x2 degrees) is exceptionally large (about 31,000), so a few dozen parameters carry very little weight. We have clarified this in the text by adding to the methods section: "where N is the number of degrees of freedom (equal to the number of observations in this case)" and including the Michalak et al. (2005) reference if readers want more information.

2. the authors state that "Because representation errors will be large at lowresolution this analysis cannot be performed using the low-resolution model used elsewhere in this analysis." (P7, L7-8). I wonder then what is the informative value of a test performed on a different configuration than that used for the rest of the study? All the more that this "representation error" directly influence the value of  $\chi_r^2$ , as it is included in the observational covariance error matrix  $C_d$ . We thank the reviewer for the comment. The reviewer is correct in that the analysis can be performed at low-resolution, we have corrected this point in the text. However, we note that it is not recommended to assess the chisquared at a low resolution considering that an actual assimilation of the data (which is future work) would not be performed at such a low resolution as representation errors can be large (i.e. not representing the heterogeneity of the land surface). The value of using this test under a different configuration is an attempt to assess the model/structural uncertainty as recommended during the review i.e. can the model reproduce the measurements? In order to assess whether the model is capable of simulating the measurements (which is what the chi-squared analysis tests), it is more informative to do so at a high spatial resolution as would be used for an assimilation of the data. To clarify this, we have amended the manuscript to explain why we apply a low-spatial resolution for the error propagation but high-resolution for the chi-squared test. We added a better description of this in the methods and added extra clarification in the discussion section.

3. More generally, it seems to me that the value of  $\chi^2_r$  should anyway expected to be larger than one. Indeed, no optimization is performed in this study, so that model-data wont likely be excellent, and the  $\chi^2_r$  value precisely translates a goodness-of-fit given the trust put in the observations. Contrarily to what is stated in that paragraph (P7, L5-11), my understanding is that having a value of  $\chi^2_r$  lower than one should not be a target as it translates overfitting and underestimation of observational error (too much trust in the observations, i.e. model outputs and/or data), all the more if the prior model-data fit is not expected to be great. Note that in (Kuppel et al., 2013),  $\chi_r^2$  was estimated using the optimised model (Eq. 6) -albeit with a slightly different formulation following (Tarantola, 2005). Thanks for the comment. There are two alternative forms for the chi-squared test: one with the prior and one with the posterior. Both are used in Kuppel et al. (2013). Under the linear assumption the value of the chi-squared test in the prior case and posterior case are mathematically identical - although their formulation is different. Therefore the issue of using the prior or posterior does not arise. One way of thinking about this is that (i) in the prior case the model-data mismatch may be large, but the parameter uncertainties are also large which results in a larger spread of model realisations, and (ii) in the posterior the model-data mismatch will be smaller, but the parameter uncertainties are also smaller which results in a smaller spread of model realisations. They therefore give a similar value (exactly the same in the linear case). We think that perhaps the reviewer has confused the  $\chi^2_r$  test with the average model-data mismatch (which should be greater than one for the prior), but which ignores the spread of model realisations; i.e. the  $HC_{x_0}H^T$  term. We have modified the methods section to better explain what the chi-squared test means. Also, Michalak et al., 2005 gives a good description of this and we have added this reference for the readers.

I would thus recommend the authors to assess the reduced chi-squared statistics at the same resolution as that used for the rest of the analysis, and discuss on this basis. The advantage is that the impacts of both neglected structural and representation errors are included here. Thank you for the suggestion. This seems to be similar to point 2 above, please refer to that response. If the value of  $\chi_r^2$  is significantly larger than one, it can arise from underestimated structural error, underestimated representation error, and/or model-data misfit. Acknowledging this from the start and for the discussion of the presented parametric uncertainty reduction would give more depth to the paper. That is a good point and we agree with this. Please refer to the response to point 3 above. To make this clearer to readers we modified the methods to better describe what the chi-squared test means.

Second is the systematic error. I have a hard time understanding how the author can propagate a "systematic error of unknown sign", for two reasons:

1. the very basis of error propagation used in this study assumes Gaussian error

that can be propagated linearly, i.e., using matrices (P3, L14-22). The fact that the sign of the systematic error is not known does not necessarily make it suitable for statistical analysis, let alone be characterized as Gaussian. Thanks for the comment. This shows that we need a better definition of what we call systematic error. The term "systematic" here refers to the fact that the error applies to multiple observations (e.g. all data points for the summer season). It does mean it is a known bias. Additionally, with our analysis of SIF over non-vegetated surfaces for January and June we are sampling the distribution of the error in the zero-level offset. We are not including the zero-level offset itself (which is a bias correction), but rather the random error associated with calculating the zero-level offset. We have clarified this in the methods section as well as providing more detailed caption note for Fig. in the Appendix to describe what it means. How are the authors including this systematic random error of  $\pm 0.1$  W.m-2.um-1.sr-1 in their propagation framework? What "seasonality" is applied to it? We thank the reviewer for this comment. Indeed we need to make it clearer in the methods how we include this error term. We have added text to the methods section explaining what the seasonality is, and how the error is added into the terms  $C_x$  and H. Note that the sign of a bias is indeed generally known and its magnitude rarely is, but that is not enough for statistical analysis (Richardson et al., 2012, pp. 175-177). This point is similar to the first two sentences of this paragraph, please refer to the response above.

2. The author claim (P9, L21-22) to perform "a sensitivity test of incorporating this systematic uncertainty into the error propagation system to indicate how an error in the zero-level offset may propagate through to uncertainty in GPP." Following my previous comment, I was curious to see how the authors applied their methodology. Thanks for this comment. As was mentioned above in response to a similar comment, we have added text to the methods section explaining what the seasonality is, and how the error is added into the terms  $C_x$  and H. However, I could not find where the results are reported. What is then the basis for asserting that "We also find that the effect of incorporating the error from the zero-level offset in the SIF observations is negligible on posterior parametric uncertainties" in the Discussion (P19, L19-20)? Thanks for the comment. There is a paragraph at the end of the results section "Parameter Uncertainties" reporting the results. We do not add any additional figures or tables for this considering the result is not significant (i.e. max of about 1%change in parameter uncertainties) and it does not form a major part of the study.

3. Later in the Discussion, the authors state that "A known systematic error in forcing variables (e.g. Boilley and Wald, 2015) cannot be considered in the present error propagation system, however, in such a case a correction to the data should be performed as it will bias carbon flux estimates". Why not apply the same framework for measurement error? As it is formulated, the methodology is imprecise and the result not reported, so it is hard to assess what the authors have effectively done. As discussed with point one, we have better defined what we mean by "systematic error" and bias. The systematic error (i.e. bias) from Boilley and Wald (2015) differs from the one we describe for the zero-level offset.

Other comments:

- This is more of personal preference, but I think Eq. (6) should read  $d_i = d_i^t + \epsilon_i + z$  In my view, this translates more clearly the fact that retrieved measurements derive from a true value affected by random and systematic errors, and not the other way around. Yes, fair point, that does seem more logical. We have amended this as suggested. We have also added that z is of the same of probabilistic form as  $\epsilon_i$ , but it applies to more than a single index point. To make this a little clearer we have also changed z to be  $\varepsilon_z$ .
- Some questions marks are present in the revised manuscript where references should be, presumably from missing LateX bibliography pointers. Ah yes, it seems it did not compile fully. This is now fixed.

[revised manuscript text omitted]